# Enhanced ventilation of Eastern North Atlantic Oxygen Minimum Zone with deglacial slowdown of Meridional Overturning

**Sofía Barragán-Montilla** [1,5] ✉, **Heather J. H. Johnstone** [1], **Stefan Mulitza** [1,2], **Dharma A. Reyes Macaya** [1,3,4], **Babette A. A. Hoogakker** [3] **& Heiko Pälike** [1,2]

The eastern Tropical North Atlantic Oxygen Minimum Zone (ETNA OMZ) plays a critical role in marine ecosystems off northwestern Africa. One of the key controls of the ETNA OMZ is ventilation driven by the subsurface ocean circulation of the Atlantic subtropical gyres. However, how this shallow circulation interacts with changes in the strength of the Atlantic Meridional Overturning Circulation (AMOC) remains unclear. Here, we present a deglacial and high-resolution paleo-oxygenation record (combined bottom and pore water) from the margin of ETNA OMZ (GeoB9512-5, 793 m water depth), which registers more strongly oxygenated periods during the Last Glacial Maximum (LGM), two parts of the Heinrich Stadial 1 (HS1), and during the Younger Dryas (YD). We show that steeper meridional temperature gradients during HS1 and YD associated with AMOC slowdown intensified the subsurface subtropical cell circulation and increased the oxygen supply to the ETNA OMZ.

Oxygen Minimum Zones (OMZ) are found beneath highly productive upwelling systems[1] that sustain unique marine ecosystems in the world oceans[2–5]. The diversity of these ecosystems depends, among other factors, on the oxygen supply, which has decreased in the last decades, partly due to global ocean warming associated with anthropogenic greenhouse gas emissions[6,7]. Another concern in a warming world, are projections that the large-scale deep circulation of the ocean, the Atlantic Meridional Overturning Circulation (AMOC), could possibly slowdown in the foreseeable future[8–12]. Although a future AMOC slowdown is still under debate, our understanding of its relationship with subsurface ventilation driven by shallow wind-driven subtropical cells (STCs) requires attention.

Since the available instrumental oxygen records at any ocean depth date only from the 1970s[13], our knowledge must rely on marine paleo-oxygenation records. One approach comes from detailed benthic foraminifera taxonomic analyses used to infer changes in ocean oxygen concentrations[14,15]. Changes in this parameter actively modify benthic organisms' physiological responses and are registered as shifts in biodiversity and distribution patterns of benthic foraminifera[5,16–18]. Benthic foraminifera are a microfossil and extant group that makes up around 50% of the eukaryotic biomass in modern oceans[19] and are one of the most diverse microorganisms in the ocean[20]. Their distribution mostly depends on seafloor food and oxygenation[21–26], and since the environmental preferences of several species are well documented, environmental models based on benthic foraminifera assemblages hold promise to investigate oceanic oxygen variability, particularly in OMZs[27–31].

In this study, we used the Enhanced Benthic Foraminifera Oxygen Index[15] (EBFOI), supplemented with additional data on benthic foraminifera oxygen preferences[14,32–35] (Supplementary Data 2–4), to reconstruct the deglacial paleo-oxygenation trends of the mid-depth

[1]MARUM—Center for Marine Environmental Sciences, University of Bremen, Bremen, Germany. [2]Faculty of Geosciences, University of Bremen, Bremen, Germany. [3]Lyell Centre, Heriot-Watt University, Edinburgh, UK. [4]Millennium Nucleus UPWELL, CEAZA - Centro de estudios en zonas áridas, La Serena, Chile. [5]Present address: Kiel University, Institute of Geosciences, Kiel, Germany. ✉e-mail: sbarraganmontilla@marum-alumni.de

OMZ lower boundary (300−800 m water depth) of the eastern tropical North Atlantic (ETNA OMZ) (300−800 m water depth). This OMZ, which reaches oxygen concentrations as low as 40 μmol/kg in its core[36–38], forms in the "shadow zone" between the north and south subtropical gyres[39], where older water masses with lower renewal rates[40] lead to poorly ventilated regions that make up the ETNA OMZ (Fig. 1a). The high productivity of the surface waters off the NW African coast is driven by upwelling induced by the northeast trade winds[41], which forms part of a shallow wind-driven overturning circulation: the Subtropical Cell (STC). The upwelled waters are replaced in the subsurface through Ekman downwelling of surface waters in the subtropical convergence[40].

To reconstruct the combined bottom and pore water oxygen (BPWO) changes of ETNA OMZ, we used the sedimentary record of gravity core GeoB9512-5, retrieved off the coast of Senegal in northwestern Africa[42] (15°20'13.20"N, 17°22'1.20"W, 793 m water depth, Fig. 1). The core is situated south of the North Atlantic subtropical gyre

in the zone of seasonal (winter) upwelling. Modern dissolved oxygen concentrations at the core site are around 88 μmol/kg[43] and correspond to the deep margin of the ETNA OMZ (Fig. 1b), which separates the lower oxygen conditions of the OMZ with the more oxygenated waters below and makes it a sensitive recorder of changes in this OMZ boundary.

The BPWO record presented here is based on a stratigraphic framework modeled with 16 Accelerator Mass Spectrometry radiocarbon dating of planktic foraminifera[44] (Fig. 2) and covers the last 27,000 years. This time interval includes periods of AMOC decline, registered in the sedimentary ²³¹Pa/²³⁰Th ratios records[45–48] during the Heinrich Stadial 1 (HS1, -17.6 – 14.7 ka BP) and Younger Dryas -12.6 – 11.8 ka BP), also seen in carbon isotope records[49] ($\delta^{13}C$). Although uncertainties exist in the ²³¹Pa/²³⁰Th proxy as it can be overprinted by, for example, opal fluxes[50], a large number of available data for the Atlantic allows drawing a reliable record of the large-scale changes seen in AMOC variability[51].

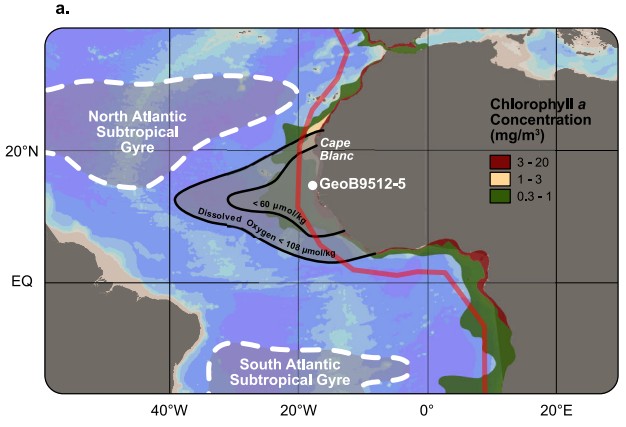

**Fig. 1 | Location of site GeoB9512-5 and the Eastern Tropical North Atlantic Oxygen Minimum Zone. a.** Geographic location of gravity core GeoB9512-5 (15°20'13.20"N/17°22'1.20"W, 793 m water depth), the Atlantic subtropical gyres and the Eastern Tropical North Atlantic Oxygen Minimum Zone outline at 500 m water depth (shadowed) are located following previous studies[37,40,98]. Upwelling areas are indicated with the Approximate Chlorophyll *a* concentration (mg/m³)

zones based on Aqua-Modis data (4 km resolution, in November 2019, https://oceancolor.gsfc.nasa.gov/); **b.** GeoB9512-5 location and modern configuration of the Eastern Tropical North Atlantic Oxygen Minimum Zone at depth (dashed outline). Plotted with Ocean Data View[99] using the GLODAP v2.2022 oxygen data base[43], interpolated by DIVA.

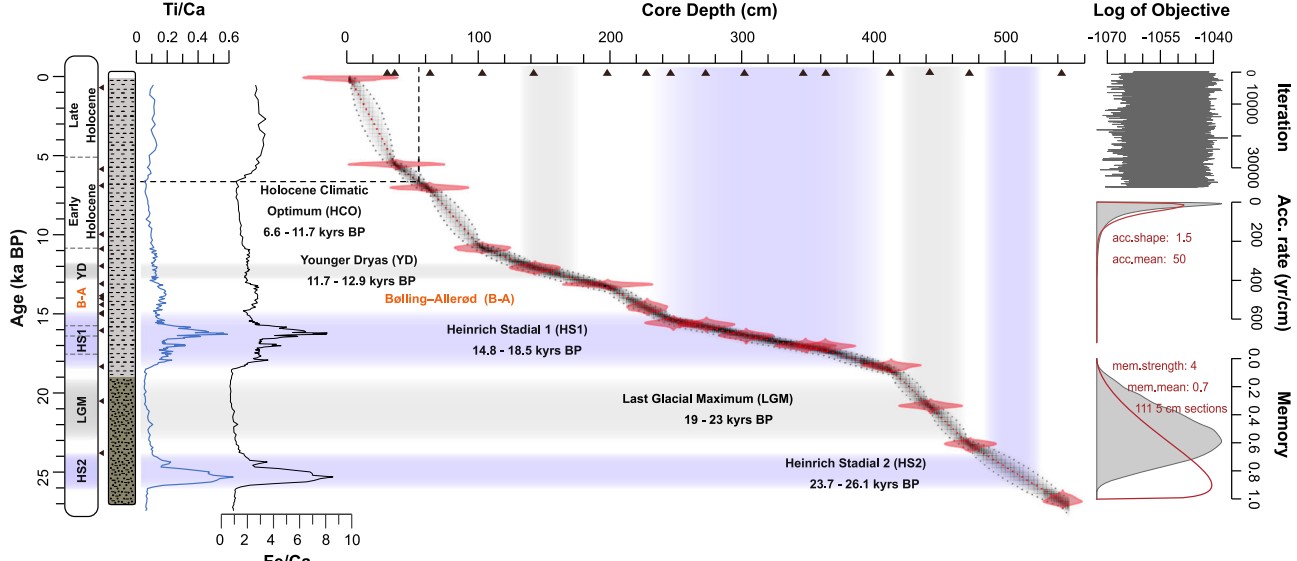

**Fig. 2 | GeoB9512-5 age model with the key climatic events in the last 27,000 years.** Downcore median calendar ages modeled using 16 radiocarbon ages (triangles in the Core Depth axis)[44] in the Bacon Package[87] with the Marine20

calibration[88]. The timing of the Heinrich Stadials is supported by elevated Fe/Ca and Ti/Ca ratios[100] representing enhanced dust deposition during continental aridity[77].

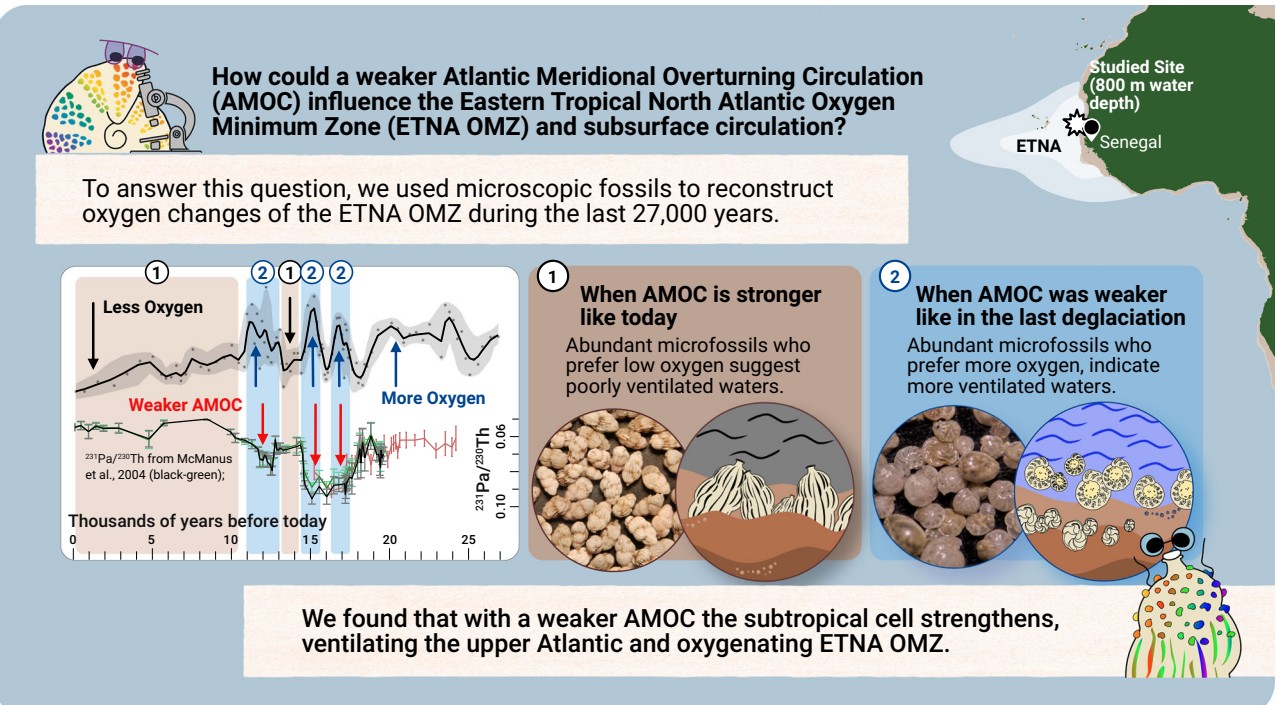

**Fig. 3 | Graphic Abstract.**

Here, we present a high-resolution continuous BPWO record of the last deglaciation in the eastern tropical North Atlantic to investigate the effect of AMOC-induced circulation changes on the STC during times of documented AMOC shoaling (LGM) and slowdown (HS1 and YD) (Fig. 3).

## Results

Our record shows bottom/pore water dissolved oxygen concentrations (BPWO in Fig. 4e) of between 92 and 257 μmol/kg (EBFOI between 17.9 and 98.7, Fig. 4e). BPWO reconstruction shows high oxygen conditions (dissolved oxygen concentrations >161–192 μmol/kg, 95% confidence interval-CI) at the ETNA OMZ margin from the onset of the record (-27 ka BP), through most of the last glacial until 18.9 ka BP. This period was interrupted only by one drop in BPWO centered on 25.1 ka BP. Even higher values (on average 180–224 μmol/kg, 95% CI) were recorded between 24 - 23.5 ka BP (Supplementary Data 1), and to a lesser extent during the LGM (169–193 μmol/kg, 95% CI). The deglaciation was characterized by abrupt transitions from oxygenated conditions to lower BPWO periods. After 18.9 ka BP, at the onset of the Heinrich Stadial 1 (HS1), BPWO concentrations decreased to an average 108–137 μmol/kg (95% CI) with lower values of 103–132 μmol/kg (95% CI) around 17.8 ka BP. This was followed by a rapid increase in BPWO during HS1, with two distinctive BPWO peaks (1) of 166–207 μmol/kg (95% CI) around 16.8 - 16.5 ka BP and (2) of 182–231 μmol/kg (95% CI) at 15.5 - 14.8 ka BP, which represents one of the highest BPWO values of the record. Between 16.2 and 15.9 ka BP, BPWO concentrations decreased again (average 113–152 μmol/kg, 95% CI).

BPWO dropped below 122–158 μmol/kg (95% CI) between 14.6 and 13.2 ka BP during the Bølling–Allerød (B-A). A transient BPWO increase (148–179 μmol/kg, 95% CI) was recorded between 13.1 ka and 12.6 ka BP during the YD, which was followed by the highest BPWO values in our record of 154–230 μmol/kg (95% CI) at 12.2 ka BP. This BPWO increase persisted at the onset of the Holocene when the final peak of 170–217 μmol/kg (95% CI) was registered between 11.3 and 10.9 ka BP. The last BPWO increase (on average 126–147 μmol/kg, 95% CI) occurred between 5 and 4.2 ka BP, and was followed by the lowest BPWO concentrations in the record of 97–101 μmol/kg (95% CI) between 1.6

and 0.2 ka BP. These values compare well with the modern measured BPWO concentration of 87.6 μmol/kg at our site[43,52].

The paleo-BPWO record presented here is characterized by an alternation of relatively low BPWO intervals and BPWO peaks (Fig. 4e), as seen by the variations of the proportions of oxic, suboxic and dysoxic species used in the EBFOI calculations (Fig. 5b). The relative abundance of stress species (Fig. 5c), which are foraminifera species better adapted to environments of reduced oxygen and increased organic matter availability[22], are consistent with these interpretations as they are present in average percentages over 50 % in lower BPWO intervals, and under 50 % in higher BPWO intervals.

In our record, a high abundance of infaunal (Fig. 5d) and stress species (Fig. 5c) indicates eutrophic and mesotrophic seafloor environments (moderate to high organic matter concentrations) throughout the whole record, from the last glacial into modern times (Supplementary Data 1). Although acids released by remineralization of organic matter under oxic conditions can result in calcite dissolution in seafloor sediments[53,54], foraminifera tests appeared well preserved throughout the core.

## Discussion

We observe an association between oxic intervals at our site (Fig. 4e) and AMOC perturbations (Fig. 4a), that aligns with observations that the major shifts in ocean circulation during these events altered ventilation throughout the whole ocean[55]. Increased BPWO is recorded during the LGM, when AMOC circulation was shallower and likely stronger than it is today[56], while the highest values of the record occur in HS1 and the YD, when AMOC circulation was severely reduced. A slight increase in AMOC strength between 5.5 - 3 ka (Fig. 4e) is also associated with a brief increase in BPWO in our record, suggesting AMOC control on ETNA OMZ oxygenation also under interglacial conditions.

Modern low oxygen conditions of the ETNA OMZ are set by the poorly ventilated Eastern South Atlantic Central Water and the southward flow of better ventilated North Atlantic Central Water[37,57] (NACW). On the other hand, cold periods in the North Atlantic are associated with southward shifts of hydrographic and atmospheric

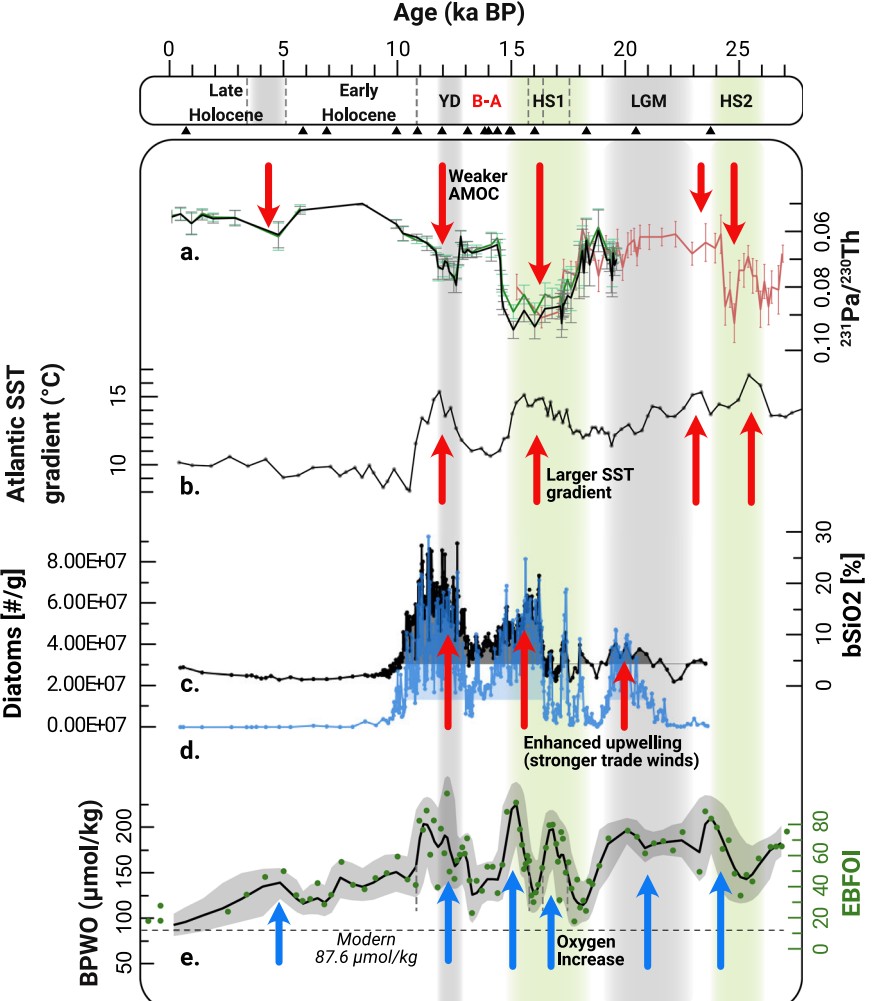

**Fig. 4 | GeoB9512-5 deglacial bottom/pore water oxygenation (BPWO) record and Atlantic paleoceanographic circulation proxies. a** Sedimentary protactinium-thorium ratio ($^{231}$Pa/$^{230}$Th) from the Bermuda Rise[47] (black and green curves), and deep western North Atlantic[45] (red curve), error bars represent 2 s.e.m. **b** Atlantic Sea Surface Temperature (SST) gradient estimated by subtracting SST at site SU8118[66] (37.76 N, 10.18 W) from SSTs at site M35003-4[67] (12.09 N, 61.243 W); NW Africa Upwelling proxies. **c** Opal (bSiO2 %). **d** Diatom content during the last deglaciation from site GeoB7926-2[80] (20°13'N, 18°27' W), shaded areas within the curve indicate enhanced upwelling periods. **e** GeoB9512-5 mean bottom and pore water oxygen (BPWO) in μmol/kg (Supplementary Data 1) calculated using the enhanced benthic foraminifera oxygen index (EBFOI, black curve), shaded area corresponds to the 95% confidence interval and gray dots represent the raw data. **Key Climate events:** Heinrich Stadial 2 (HS2); Last Glacial Maximum (LGM); Heinrich Stadial 1 (HS1); Bølling–Allerød (B-A); Younger Dryas (YD). Triangles in the age axis indicate radiocarbon ages.

frontal systems[58–60]. Southward movement of the front between northern and southern sourced central waters allowed oxygen-rich NACW to penetrate further south along the NW African margin[61]. At the same time, displacement of the major wind systems and the ITCZ[62] lead to the North Atlantic subtropical gyre being positioned several degrees south of its current location during the LGM, HS1 and YD[63]. This would have exposed the core site to the younger and better ventilated waters of the North Atlantic Subtropical Gyre and is also supported by $\Delta CO_3^{-2}$ records of the west Atlantic that show low calcite saturation related to increasing better ventilated northern sourced waters during the HS1 and YD[64]. During the B-A and Holocene times, as AMOC deepened and strengthened (Fig. 4a), the subtropical gyre moved north, contributing to reduced BPWO conditions registered by benthic foraminifera in our site (Fig. 4e).

The North Atlantic sea level pressure gradient and hence the anticyclonic wind circulation driving the STG is also tightly related to the distribution of sea surface temperature[65]. To illustrate the relation between North Atlantic Sea Surface Temperature (SST) and OMZ ventilation, we estimated the deglacial SST gradient across the Atlantic by calculating the difference in SST between site SU81-18[66] in the northeastern STG and site M35003-4[67] in the tropical west Atlantic using interpolated datasets (Fig. 4b). Modest surface cooling in the tropical west Atlantic, and greater cooling in the northeast Atlantic during the LGM, increased the temperature gradient compared to the Holocene. Increases in the meridional temperature gradient and wind stress were even more intense during HS1 and YD (Fig. 4b) when reduction in the upper branch of AMOC warmed the sea surface and upper tropical west Atlantic[67–70], while input of glacial meltwater reduced SST in the northeast of the gyre[66]. Stronger winds would have increased subduction of surface waters into the subtropical cell, hence ventilating the upper ocean.

Such a higher SST gradient coincides with increased BPWO in the GeoB9512-5 record during the last glacial (Fig. 4e). This is related to increased glacial wind stress curl that deepened the Subtropical Gyre circulation, ventilating the subsurface and intermediate Atlantic[71]. In addition to ventilation by stronger winds, lower nutrient input, due to glacial stratification of the Southern Ocean[72], may have reduced oxygen consumption in the upper ocean as a whole. This is supported by

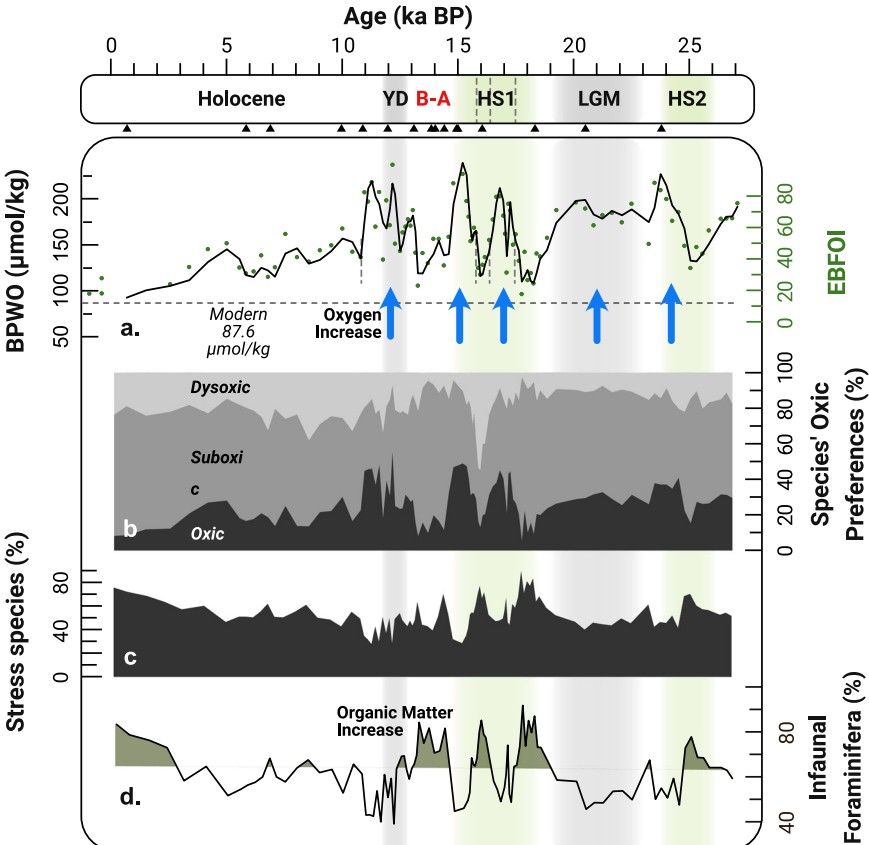

**Fig. 5 | *GeoB9512-5 downcore benthic foraminifera paleoenvironmental indicators.* a** Two-point moving average bottom and pore water oxygen (BPWO μmol/kg, black curve) estimated from the Enhanced Benthic Foraminifera Oxygen Index (EBFOI, green dots). **b** Relative abundances of Oxic, Suboxic and Dysoxic species. **c** Relative abundances of Stress Species; and (**d**) infaunal benthic foraminifera. **Key Climate events:** Heinrich Stadial 2 (HS2); Last Glacial Maximum (LGM); Heinrich Stadial 1 (HS1); Bølling–Allerød (B-A); Younger Dryas (YD). Triangles in the age axis indicate radiocarbon ages.

increasing BPWO conditions in the LGM compared to the Holocene, recorded in sites of the East Pacific and tropical Atlantic above 1000 m[73] (Supplementary Fig. S3.1b in Supplementary Information 3) and the mid-depth/upper Atlantic[74,75] (Supplementary Fig. S3.1a in Supplementary Information 3), while the deep ocean was low in dissolved oxygen and high in carbon[73,76].

The high temperature gradients across the North Atlantic sea surface registered in times of AMOC slowdown during HS1 and the YD (Fig. 4b) coincide with the highest BPWO values of the GeoB9512-5 record (Fig. 4e). Strong northeasterly winds over NW Africa[77] enhanced the upwelling off the African coast which, combined with increased nutrients in the upper Atlantic[78], strengthened productivity as shown in diatom and opal flux records (Fig. 4c, d) from the West African coast[41,79,80]. In our record, BPWO is high in two distinct oxygen peaks during HS1 and during the entire YD. Although high productivity may have led to more oxygen consumption in the water column[81], increased ventilation from stronger winds in the formation area of North Atlantic Central Waters was evidently the dominant factor controlling the BWPO at our site.

Increased BPWO during HS1 has also been recorded in the upper West Atlantic[32], which in our deglacial BPWO record is observed as a pronounced double peak, a pattern that is absent in the deep [231]Pa/[230]Th record from the Bermuda Rise[45,47] (Fig. 4a). A recent compilation of [231]Pa/[230]Th records from the tropical and North Atlantic[48] indicates two phases of AMOC slowdown, an early slowdown due to meltwater input and iceberg calving from the Eurasian Ice Sheets from about 19 to 16.5 ka BP, and a later slowdown associated with Laurentide iceberg calving from about 16.5 to 15 ka BP. Two phases of ice rafting and meltwater input are also recorded in Atlantic sediments[82].

Whether the transition between the two phases of meltwater input is associated with a continuous AMOC decline seems currently unclear. Deglacial radiocarbon records from the Greenland Sea indeed show a brief phase of stronger vertical convection from about 15.9 to 15.2 ka[83]. We thus suggest that the two phases of AMOC slowdown expressed in ETNA OMZ ventilation as a double peak in BPWO at GeoB9512-5 (Fig. 4e), were likely interrupted by a brief AMOC intensification, which might not be resolved in deglacial core sections, or is below the detection threshold of [231]Pa/[230]Th.

Our data shows that a decline in AMOC strength increased ventilation of the Eastern Tropical Atlantic Oxygen Minimum Zone. This is despite AMOC perturbations being associated with an increase in temperature and salinity of the upper (< 1500 m) tropical Atlantic[63,69,70,84], which would act to reduce oxygen solubility - and, during HS1 and the YD, an increase in productivity off West Africa[85] - which would increase oxygen consumption. Our findings imply that the future of the ETNA OMZ is critically dependent on the state of the AMOC, and that any future decline might counteract the current deoxygenation trend related to global warming.

## Methods
### Chronology and Age model
The downcore ages from site GeoB9512-5 (Fig. 2) were modeled using 16 Accelerator Mass Spectrometry (AMS) radiocarbon ages from planktic foraminifera (*Trilobatus sacculifer* and *Globigerina bulloides*) samples[44]. The samples were measured at the MICADAS laboratory at the Alfred Wegener Institute (AWI)-Bremerhaven[86]. A continuous age model was calculated with the R script BACON[87] version 2.5.5 and the Marine20 calibration curve[88] to obtain a median calendar age and

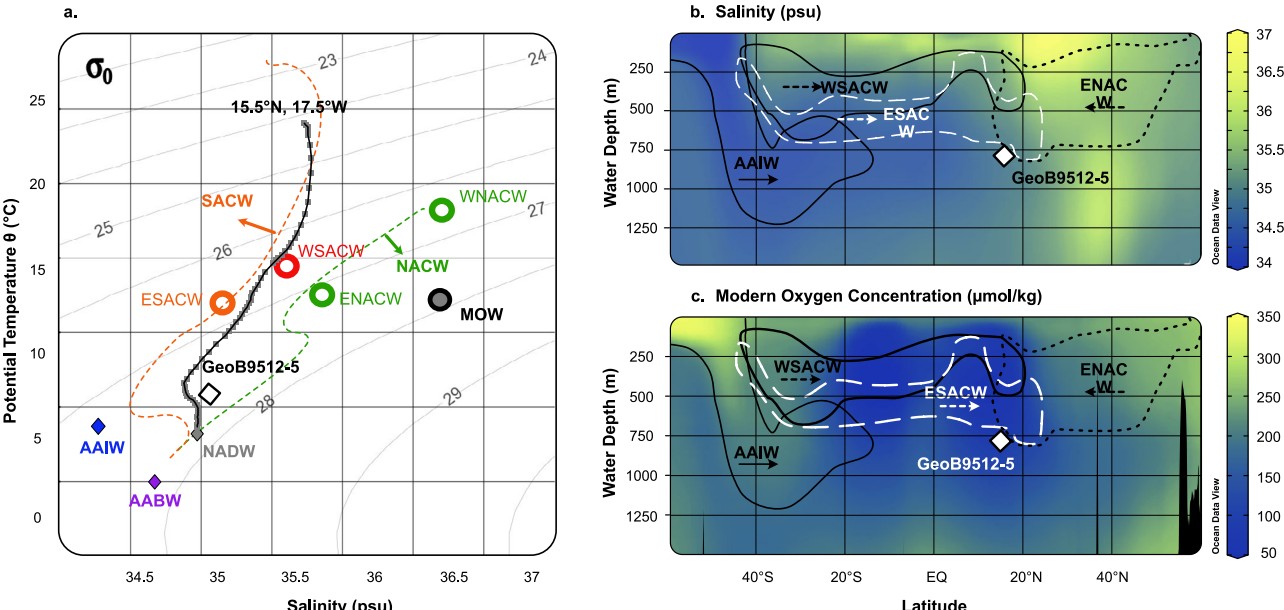

**Fig. 6 | Central and Intermediate water masses in the studied site GeoB9512-5 (15°20′13.20″N, 17°22′1.20″W, 793 m water depth). a** Temperature-Salinity diagram showing the water masses sources in the eastern North Atlantic, adapted from previous studies[40,57], dashed lines represent the transition of North and South Atlantic Central Waters (NACW and SACW, respectively) in the eastern North Atlantic[85]. **b** Salinity showing the eastern North Atlantic central water masses distribution in the studied area[40]. **c** Dissolved oxygen concentration (umol/kg)[43] and modern extension at depth of the East North Atlantic Oxygen Minimum Zone (ETNA). Oceanographic sections (a and b) were plotted with Ocean Data View[99] using the GLODAP v2.2022 oxygen data base[43], interpolated by DIVA.

uncertainty for each sampled depth. The resulting age model shows a deglacial sequence deposited in the last 27,000 years with no age reversals and includes the key climatic periods Heinrich Stadial 2 (HS2), Last Glacial Maximum (LGM), Heinrich Stadial 1 (HS1), Bølling−Allerød warming (B-A), Younger Dryas (YD), and the Holocene Climatic Optimum (HCO), that divides the early Holocene from the Late Holocene (Fig. 2).

### Oceanographic setting

Water masses in our site were identified using a T-S diagram showing GeoB9512-5 situation (Fig. 6a), along with the existing available references. The studied gravity core GeoB9512-5[42] (15°20′13.20″N, 17°22′1.20″W, 793 m water depth, Fig. 1), is situated in the "shadow zone" between the north and south Atlantic subtropical gyres (Fig. 1a). At our site, modern low oxygen values of 88 µmol/kg characterize the ETNA OMZ lower margin (Fig. 1b), and bottom water temperatures and salinities are approximately 6.9 °C, and ~34.8 (Fig. 6a) respectively[43]. Such conditions position GeoB9512-5 in the mixing area of the North Atlantic Central Water (NACW) and Eastern South Atlantic Central Water[40] (ESACW, Fig. 6).

The South Atlantic Central Water flows northward from the South Atlantic subtropical gyre[57], where the coincidence of wind-driven downwelling and seasonal buoyancy drives waters into the thermocline[89]. In the Cape Verde area, this water mass is present as Western and Eastern South Atlantic Central Waters (Fig. 6b, WSACW and ESACW respectively). The WSACW is differentiated by its formation area in the western South Atlantic, and even though it is mainly concentrated in the western Atlantic it reaches the eastern North Atlantic to some extent[57]. The ESACW is a nutrient-rich water mass formed at the mixing area of the Agulhas Current and the South Atlantic Current south of Africa[40,57]. Above the SACW masses, the North Atlantic Central Water (NACW) flows southward and is represented in our area by the Eastern North Atlantic Central Water (ENACW, Fig. 6b, c), bringing –low nutrient, better ventilated waters from its formation area, the inter-gyre region (between 39-48°N)[57].

The interaction of these water masses plays an important role in the Eastern Tropical North Atlantic Oxygen Minimum Zone (ETNA OMZ), as the northward flow of poorly ventilated and lower oxygenated waters of the ESACW are counteracted by the southward flow of the oxygenated ENACW[37,57]. The site studied here (GeoB9512-5) is located in the lower ETNA OMZ in a mixture of central waters, where more than 50% of the lower OMZ is NACW which has retroflected in the gyre, at around 14°N[90].

### Benthic foraminifera taxonomy and quantitative analyses

Sediment samples were (1) washed through a 63 µm sieve with deionized water; (2) dried in an oven at ~45 °C for no more than 24 hours; and (3) dry sieved through 63, 125, 150, and 250 µm, and stored and labeled in glass vials. The >150 µm fraction was analyzed to extract benthic foraminifera (for taxonomic and quantitative analyses) and planktic foraminifera (for radiocarbon dating). Paleoenvironmental and paleo-oxygenation results and interpretation are based on the taxonomical identification and the quantitative analyses of at least 200–250 benthic foraminifera from 100 samples from core GeoB9512-5[91].

Benthic foraminifera were morphologically separated, and the taxonomical identification was made to species level in most cases (Supplementary Data 2). Genera were determined following Loeblich and Tappan (1987)[92], and species were identified based on multiple refs.[93–95]. The updated taxonomy was finally revised using the online database WoRMS[96] (Supplementary Data 2). Digital images of the benthic foraminifera extracted during this study were taken at the Microscopy Laboratory at MARUM, using a Keyence VHX 6000 digital microscope with a motorized stage (Supplementary Information 1).

### Bottom/Pore Water Oxygenation reconstruction and uncertainties

The BPWO record presented here was based on the Enhanced Benthic Foraminifera Oxygen Index[15] (EBFOI). The terms high oxic (>3 ml/l (>131 µmol/kg)), low oxic (1.5 – 3 ml/l (65 – 131 µmol/kg),

suboxic (0.3 – 1.5 ml/l (13. – 65 μmol/kg)), dysoxic (0.1 – 0.3 ml/l (4 – 13 μmol/kg)) and anoxic (<0.1 ml/l (<4 μmol/kg)) are used here as previously defined[15]. EBFOI relies on multiple species, and of the 151 taxonomic units identified at our site, 143 (including all abundant species) have information about their oxygen preference (Supplementary Data 2 and 4). Although the EBFOI is one of the best tools to quantify the dissolved oxygen concentration changes from the whole livable habitat[15] (bottom and pore water oxygen concentration), it can be limited by the availability of information about the oxygen preferences of benthic foraminifera species and, has been shown to underestimate oxygen concentrations[32]. For this study, we used the existing databases of benthic foraminifera oxygen preferences[14,15,32] and supplemented this with published information on living assemblages from seafloor-surface samples from the eastern tropical Atlantic[33–35] (Supplementary Fig. S2.1 in Supplementary Information 2). The methodology is described in detail in Supplementary Information 2 and resulted in an updated compilation ("(1) Total Compilation" in Supplementary Data 3), that was used in the transfer function of Kranner et al.[15] to calculate dissolved oxygen concentrations in ml/l.

To facilitate the interpretation of our record in contrast to already published paleo-oxygenation data, and as recent literature includes oxygen descriptions in μmol/kg, we converted our ml/l BPWO record to μmol/kg. Dissolved oxygen concentrations in ml/l for each sample from site GeoB9512-5 (Supplementary Data 1) were converted to μmol/kg by multiplying the calculated oxygen (ml/l) by the molar volume of oxygen (44.66 μmol/ml) and then dividing by the bottom water density of site GeoB9512-5 (1.027 kg/l). The density was computed using the specific bottom water salinity and temperature at our site, estimated from GLODAP version 2.2022[43] in the seawater state equation[97]. The universal NOAA/WOD18 notes conversion (1 ml/l of oxygen is equivalent to 43.570 μmol/kg) was also reported for reference.

We also assessed the accuracy of our reconstruction by comparing our paleo-oxygenation record with downcore relative abundances of stress species (Fig. 5c), that are better adapted to oxygen-depleted environments in organic matter rich-environments (*Bulimina, Bolivina, Cassidulina, Melonis, Fissurina, Globobobulimna*)[26].

To estimate the uncertainty associated with our data, we assumed that oceanographic conditions, including AMOC strength, were relatively stable in samples younger than 2.6 ka BP ($n = 6$), allowing us to estimate reproducibility. Dissolved oxygen calculated from EBFOI for four samples (between 2.5 and 17.5 cm) in gravity core GeoB9512-5 and two samples (2 cm, 8 cm sediment depth) from nearby multicore GeoB9512-4[42] (Supplementary Data 2) gave a mean value of 96 μmol/kg, close to the modern 88 μmol/kg[43], and standard deviation of 7 μmol/kg. Since the distribution of our data is normal ($p$-value = 0.114 using the Shapiro test), the 95% confidence interval (CI) was estimated considering the age uncertainty and the standard deviation reported here (Fig. 4e[1]), combined with 10,000 downcore proxy series produced with the Bacon script for R[87]. The mean oxygen values are detailed in Supplementary Data 1.

## Data availability

The benthic foraminifera counts - taxonomical information and Accelerator Mass Spectrometry Carbon 14 data used in this study are available in the PANGAEA® - Data Publisher for Earth & Environmental Science, with the identifiers [https://doi.org/10.1594/PANGAEA.962951] and [https://doi.org/10.1594/PANGAEA.962899]. The data generated in this study have been deposited in the Zenodo repository with the identifier [https://doi.org/10.5281/zenodo.10724771]. The paleoenvironmental information per sample, benthic foraminifera oxygen preference data, and the benthic foraminifera images generated in this study are also provided in Supplementary Data 1, 2 and Supplementary Information 1, respectively.

## Code availability

The code used to process and visualize the benthic foraminifera data can be found at Barragán-Montilla (2024) via the identifier https://doi.org/10.5281/zenodo.10724771. The script for the age model is based on Blaauw and Christen (2011) and processing and visualization of the data is available in the same repository.

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

## Acknowledgements

This research was funded through the Cluster of Excellence "The Ocean Floor – Earth's Uncharted Interface" (EXC-2077, Project 390741603). The GeoB Core Repository at MARUM provided the sample material used here. The authors thank Master students Jakob Quabeck, who produced some of the microfossil images and Ishani Rathnayake, who helped with sample preparation. This research is also supported by GLOMAR – Bremen International Graduate School for Marine Sciences, University of Bremen.

## Author contributions

S.B.M., H.J.H.J., S.M., D.A.R.M., B.H.A.A. and H.P. contributed to the preparation of the final manuscript. S.B.M contributed with the conceptualization, core sampling, sample preparation, benthic foraminifera extraction, taxonomy and quantitative analyses, data visualization, interpretation and discussion, and wrote the manuscript; H.J.H.J. contributed with the conceptualization, data interpretation and discussion, and manuscript editing; S.M. contributed with the core sampling, conceptualization, age model, discussion, and manuscript editing; D.A.R.M. and B.H.A.A. contributed with the quantitative analyses, oceanographic data revision, and manuscript editing; H.P. contributed with the conceptualization, data discussion and manuscript editing.

## Funding

## Competing interests

The authors declare no competing interests.
