## [Peer Review file · Nature Communications]

Enhanced ventilation of Eastern North Atlantic Oxygen Minimum Zone with deglacial slowdown of Meridional Overturning

Corresponding Author: Dr Sofía Barragán-Montilla

Version 0:

Reviewer comments:

Reviewer #1

(Remarks to the Author)

This paper by Barragan-Montilla et al., presents evidence from benthic foraminifera assemblages that oxygen increased at shallow-intermediate depths in the ETNA associated with periods of decreased AMOC strength during the last deglaciation. The record and interpretation are sound and paper is well-written and well organized. Overall, this is an interesting finding with potential significance for understanding how Atlantic oxygenation could respond to future climate perturbations.

My primary critique is that the potential impact of this contribution is limited by a lack of context. I'd suggest expansion upon the current scope in two ways:

- 1) Including a more indepth discussion of the link between AMOC and ventilation in this region. Are there records from other regions of the Atlantic that could test the author's assertion of ventilation changes? How about the mechanism in terms of northward gyre shift? Given the potential significance for future climate the mechanic piece linking oxygenation in the ETNA and AMOC strength is particularly important
- 2) The deglacial Atlantic is probably one of the best sampled times and regions in paleoceanography. Thus, it's a bit of a missed opportunity that this paper is not more in conversation with other records. This should clearly include other regional paleo-oxygen records. For example, the authors mention in passing a contrast with deeper oxygenation (Lines 149-151). Exactly where and at what depth these records are important, and should be discussed if not shown in a summary figure. Beyond oxygenation, though, both carbonate chemistry and temperature records from the ETNA and North Atlantic more broadly should be used to contextualize these findings.

Minor things

- I'd strongly suggest the authors avoid the terms "low oxie" and "high oxie". Ocean oxygen research is already plagued by too many poorly defined or inconsistent terminology. Neither of these terms are in common use and introducing additional terminology here is not a service to the pape. As a quantitative oxygen record is generated, I'd recommend using actual quantitative values throughout. If the authors must use a qualitative term for oxygen range, then this range (not just the 3ml/L divide between them) needs to be defined in the body at first use and not in the methods.

- Line 131: "abundance of"

- Line 206: are -> and

(Remarks on code availability)

Reviewer #2

(Remarks to the Author)

The manuscript presented by Barragán-Montilla et al. reconstructs bottom water oxygen concentrations in the OMZ of Senegal using an index that quantifies environmental oxygen based on the oxygen preferences of benthic foraminifera. The information derived from benthic foraminifera is later interpreted in the context of AMOC perturbations and changes in the

ventilation of intermediate waters. While the micropaleontological data and age model are robust (see other comments), I have some concerns about the oceanographic context as stated in the text (see main comment #1), data interpretation (main comment #2), and the use of EBFOI as a proxy (main comment #3).

I have organised the review by first expanding on the three main comments (#1, #2, #3) and later referring to a list of other comments regarding data, figures, etc.

Comment #1: Oxygen categories, oxygen units and the OMZ of the Eastern Atlantic

Oxygen categories and the OMZ of the Eastern Atlantic: oxygen categories are not universal and thresholds vary between authors. This is commented in one of the papers cited in the text (Karstensen) but there is more literature discussing this (e.g., Paulmier and Ruiz-Pino, 2009; Hofmann et al., 2011; Levin, 2018). This is relevant in the context of this manuscript and I believe it deserves further explanation in the text.

For example, the definition of OMZ provided by Levin (2018) as areas with oxygen values <22 micromol/kg (0.5 ml/l) excludes the study area as OMZ region. This has been mentioned in Karstensen et al. (2008) and Paulmier and Ruiz-Pino (2009). Therefore, lines 283-284 "modern low oxygen values of 2.01 ml/l are typical of the OMZ margin" and considerations along the text regarding the study area as "OMZ", it would require some re-consideration/discussion. The value 2.01 ml/l is not even within the "suboxic range" indicated in line 344. For some authors it could be considered only mild hypoxia (Hofmann et al., 2011) or hypoxia (Levin, 2018).

Units: Following the information WOA18_oxygen (referenced by authors: <https://www.nodc.noaa.gov/OC5/WOD/wod18-notes.html>), oxygen units are provided micromol/kg. Most oceanographic literature discussing OMZ regions (e.g., Karstensen, Stramma, cited in the text) provide data in micromol/kg, so I wonder why authors are providing oceanographic sections in ml/l? (Figure 1).

Comment #2: Reconstruction of oxygen in bottom waters

I understand from the authors' text that the manuscript intends to reconstruct oxygen in bottom waters (lines 59-61, 199, 223). However, benthic foraminifera depend on oxygen concentrations in pore waters as well as food resources (I am not providing references as I understand the authors know them). Pore waters can have much lower oxygen concentrations than bottom waters in mesotrophic and eutrophic environments. Given the high productivity of the study area (Figure 1), it would be reasonable to expect that the supply of organic carbon to the seafloor and its degradation in the sediment would reduce the oxygen concentration in the pore waters relative to the bottom waters. Indeed, the background value of infaunal benthic foraminifera is high 40% (Figure 6). What is more, the use of "stress species" (line 350 "well adapted to oxygen depleted environments") would not be an indication of bottom water only, but of the combination of both, bottom water and pore water (and food resources). This would be even more important when the core is at the boundary of a non-OMZ region (see comment #1).

I think it would be reasonable to try to obtain additional proxies that would allow the micropaleontological signals to be interpreted with greater confidence. Sedimentary geochemical proxies (Corg, U, Mn, etc.) would be one option to try to assess the influence of primary production and independent information about the oxygenation of the bottom waters. For example, lines 137-139 refer to some previously published primary productivity inferences, but they are not in the same core and they are not plotted alongside the core being studied.

I would like to mention that, as discussed in Kranner et al. (2022), EBFOI intends to "include bottom water oxygenation (BWO) and pore water oxygenation (PWO)". The index does not refer to BWO and the oceanographic context does not allow such a direct interpretation (see comment #3).

Comment #3: The use of EBFOI as a proxy

The EBFOI index (Kranner et al., 2022) is an improved version of the Kaiho BFOI index (1994, 1999). The index has several limitations which are not mentioned in the text. I think those limitations require some reflection.

EBFOI and BFOI are based on the relative contribution of taxa stacked into oxygen categories. This means that certain species are assumed to live only in the dysoxic, suboxic, low oxic and high oxygen categories (ranges are given in the manuscript). Kranner et al. (2022) increased the number of samples from Kaiho's original 80 to 270. Despite the increase in the number of samples, this does not appear to be a large number for a worldwide ecological distribution. The original data do not follow FAIR principles, so it is not possible to reproduce or evaluate the context by which a species is assigned to an oxygen category. As far as I can tell from the papers cited above, there is no disclosure of the number of samples, type of assemblage (live, dead), ranges of variability, or optima of tolerance used to assign a species to a particular oxygen range or the impact of different size fraction in the index calculation.

For example, after applying the index, the supplementary data presented in the manuscript include assignments of oxygen levels to individuals identified to genus level (genus sp.), dubious assignments; "aff", individuals of the order Lagenina (oolina, lagenina, lavidentalina) or rare species. I wonder about the ecological/statistical basis for assigning individuals of which we do not know the species, or cosmopolitan "lagenids", rare species, to an oxygen category?

Another consideration is the maximum oxygen value obtained from the index which is around 5ml/l. If we convert ml/l to micromol/kg (<https://shorturl.at/fluGU>), we get 220 micromol/kg, which is roughly the oxygen value of bottom waters below 1250 metres (https://www.ewoce.org/gallery/A6_OXYGEN.gif). Does the value obtained from the index make sense in the context of the current/past oceanographic context (see comment #1)?

The dataset: <https://doi.pangaea.de/10.1594/PANGAEA.962951> (can only comment on the metadata, see comment below on data availability below), include, as far as understand, unpublished core tops. I wonder why authors have "reported microhabitats and oxygen conditions for most species were extracted from various references including Murray (1991),

Murray (2006) Kaiho (1994) and Kranner et al. (2022) among others” instead of using their own regional compilation and interpretation. For example, they could include not only their new core tops but also previously published datasets [I found for example: Haake, F W (1980): Relative abundance of dead benthic foraminifera in East Atlantic surface sediments off Senegal and Gambia (Table 2). PANGAEA, <https://doi.org/10.1594/PANGAEA.536233> also in Senegal, is this is not relevant?] and perhaps additional datasets from the west coast of Africa. They could compare distributions with bottom water oxygen values, with organic carbon and any other relevant environmental variables available to extract they own interpretation, which might fit or might not with the results of EBFOI. I view that suggestion as a reasonable exercise not only to support their interpretation instead of relying on previously regional data and/or interpretations from distant areas. The interpretation of benthic foraminifera as proxies for food and oxygen concentrations are not written in stone; they should be based on a reasonable set of data that allow for quantitative relations, and should be continuously updated as benthic foraminifera ecological information increases.

Other comments

Taxonomic concepts: I would like to mention that I really appreciate that authors provide images and taxonomic concepts of the identified foraminifera.

By scrolling down the list of species (supplementary table), some species designations call my attention. It appears that the taxonomy is not fully updated, as it is indicated in line 335. These are random examples (I have not checked all).

Virgulina subquamosa, unaccepted: <https://www.marinespecies.org/aphia.php?p=taxdetails&id=710270>

Nonion fabum, unaccepted:

<https://www.marinespecies.org/aphia.php?p=taxdetails&id=484789>

Eggerelloides scaber?, not recognized, *Eggerelloides scabrum* instead?

<https://www.marinespecies.org/aphia.php?p=taxdetails&id=113938>

I suggest authors access WoRMS using “worrsm” package in R (Chamberlain and Vanhoorne, 2023) to get quick access to updated taxonomy.

Transport:

Have the authors considered the influence of downslope transport in their assemblages?.

E. scabrum, *C. williamsoni*, *E. macellum* are generally “shallow water species” and *B. spatulata/dilatata/subaeranensis* and *N. fabum* are common in “shelf” environments.

I think it would be worth evaluating transport and consequently, do not include transported species in interpretations.

Datasets for review: It is impossible to review and comment on datasets (benthic foraminifera, including oxygen assignments, age model), because they are not available, they are password protected. I would like to notice that, if requested, Pangea facilitates temporary access to dataset for confidential reviewing. I have tried to convert the supplementary file with data counts to excel but the format of the table is unfriendly and it takes time to arrange columns properly for data wrangle.

Age model: Lines 257-261 and 272-273 Age model: Please clarify “median calendar ages are calculated from uncalibrated Fe/Ca and Ti/Ca” and “Median calendar ages were also calculated for sedimentary and uncalibrated”. I understand authors are using Fe/Ca peaks in their record (that have been previously shown to function as good proxies for Heinrich events) to gather additional “tie points” for age model construction. I am sorry, but I do not understand how Fe/Ca is transferred to “calibrated” years?.

It good be useful to see how the inclusion of Fe/Ca tie points fits with the radiocarbon dates/calibrated ages, in particular during HS1 when both data are available (figure 4). See comment “Datasets for review”.

Paleoclimate and paleoceanographic discussion: The discussion of lines 153-176 might benefit from a cartoon or similar depicting the paleoceanographic scenarios mentioned in the text.

Lines 165-167: Previous studies in the area indicated that enhanced easterly winds during the YD and HS1 enhanced primary production and organic carbon flux to the sea floor (Bradtmiller et al., 2016), Is then possible to conciliate enhanced productivity with increased ventilation of the thermocline?. Please refer to comment #2.

Figures:

Figure 1: shadowing obscures evaluating the vertical oxygen distribution and it does not allow to see the contouring. I suggest providing data following international oceanography convention (see comment #1) without shadowing. Once contours are provided, readers can visually identify areas of low oxygen and its value

Figure 2: how the gradient has been constructed in b) (any age interpolation has been used to subtract a record from other)?

Figure 3: label of y-axis in panel a) does not match Figure caption (BWOx versus Fisher alpha). Line in c) does not match figure caption (dissolved oxygen?).

Perhaps figure 2 and figure 3 could be merged into one.

Figure 5: I have similar considerations to Figure 1. Colouring and shadowing pre-conditions the readers to authors’ water masses identifications. Perhaps arrows, lines, etc it would be sufficient.

Line 51: "hard-shelled": consider that there are foraminifera with organic and quitinous walls when referring to foraminifera

References:

Please, re-visit reference list; the first entry has not author, sometimes journals are abbreviated and some others they are not, some references are with et al. and others are not.

References in this review:

Bradtmilller, L. I., D. McGee, M. Awalt, J. Evers, H. Yerxa, C. W. Kinsley, and P. B. deMenocal (2016), Changes in biological productivity along the northwest African margin over the past 20,000 years, *Paleoceanography*, 31, 185–202, doi:10.1002/2015PA002862

Chamberlain S, Vanhoorne. B (2023). `_worrms: World Register of Marine Species (WoRMS) Client_`. R package version 0.4.3, <https://CRAN.R-project.org/package=worrms>

Hofmann, A.F., Peltzer, E.T., Walz, P.M., Brewer, P.G. 2011. Hypoxia by degrees: Establishing definitions for a changing ocean. *Deep-sea research. Part I, Oceanographic research papers*, 58, 1212-1226. doi: 10.1016/j.dsr.2011.09.004.

Levin, L.A. 2018. Manifestation, Drivers, and Emergence of Open Ocean Deoxygenation. *Annual review of marine science*, 10, 229-260. doi: 10.1146/annurev-marine-121916-063359.

Paulmier, A., Ruiz-Pino, D. 2009. Oxygen minimum zones (OMZs) in the modern ocean. *Progress in Oceanography*, 80, 113-128. doi: 10.1016/j.pocean.2008.08.001.

(Remarks on code availability)

Reviewer #3

(Remarks to the Author)

Review on NCOMMS-24-15008

Enhanced ventilation of Eastern North Atlantic Oxygen Minimum Zone with deglacial slowdown of Meridional Overturning

The authors reconstruct the oxygen concentration in the eastern tropical North Atlantic Oxygen Minimum zone (ETNA OMZ) over the last 27 year by the use of benthic foraminifera assemblage data from a sediment core collected at roughly 800 m water depth. The authors find an oxygen increase in the ETNA- OMZ during deglacial cold events HS1 and YD. The periods of high oxygenation align well with periods of a published reduction in the Atlantic Meridional Overturning Circulation (AMOC).

The resolution of the records allows the identification of substructure in H1 and YD with several short-term oxygenation events that coincide with deglacial changes in the positions and dynamics of the North Atlantic Subtropical and subpolar Gyre that are closely linked to changes in AMOC.

The work has high significance to the field, as it allows first estimations of the coupling between northward heat transport and the vertical extension of the ETNA OMZ. The data adds valuable information in potential consequences of the current global warming.

The manuscript is well written and easy to follow. The authors use the established literature well. The conclusions and claims are supported by data. The methodology is sound, allows reproduction of the work and meets the expected standards in fields. All figures clear and self-explaining.

I see two minor weaknesses in the argumentation of the manuscript:

1. The authors use the Pa/Th record from the Bermuda rise as reference for AMOC strength. Though this record is strikingly matching the presented new results from the ETNA-OMZ, care needs to be taken with this Pa/Th record as a record for AMOC strength. The latter currently is debated to potentially be overprinted by other effects than AMOC strength. Here, more careful writing and/or addition of other proxies than Pa/Th could help to improve the manuscript.

2. The argument that increasing ENACW strength and additionally increased ventilation of ENACW causes the oxygen increase during cold periods is conclusive. Yet the authors leave out the possibility of increased ventilation and contribution of the southern sourced waters WSACW and ESACW. This probably could be added in 1 or 2 sentences.

Further Minor suggestions:

Please check the consistent use of the abbreviation ETNA OMZ sometimes OMZ or ETNA is missing.

Further minor suggestions and corrections are added to the pdf file.

(Remarks on code availability)

Reviewer #4

(Remarks to the Author)

Review of "Enhanced ventilation of Eastern North Atlantic Oxygen Minimum Zone with deglacial slowdown of Meridional

Overturning" by Barragan-Montilla et al.

The authors reconstruct oxygen concentrations in the North Atlantic OMZ over the past 20,000 years using benthic foraminiferal assemblages. This is an important contribution as there are few records in this region despite its poorly constrained future projections. They find that the OMZ becomes more oxygenated during millennial scale events (HS1 and YD) when the AMOC was likely slower. They instead link these oxygenation events to better ventilation of subtropical mode waters. Overall this is a comprehensive dataset at a unique location, and I believe it is appropriate for Nature Communications. I do have several general suggestions, followed by more specific comments.

First, throughout the manuscript, oxygen concentrations are reported as ml/l. This is likely a result of the original relationships established between benthic foraminiferal assemblages and oxygen as presented by, e.g., Kaiho 1994. However, the modern convention is to report oxygen concentrations in $\mu\text{mol/kg}$, as exemplified by data products like WOA18, GLODAPv2, and WOCE all using $\mu\text{mol/kg}$. It is rather tedious for the reader to be constantly doing the conversion throughout the text. Other papers on benthic foram assemblages, like Erdem 2019 and 2020, are starting to make the switch to these more modern preferred units, and I recommend the authors of this manuscript do so as well. Using $\mu\text{mol/kg}$ will also make it much easier to compare these results with those published using other proxies (e.g., the ^{13}C of Hoogakker et al., 2015, 2018).

Next, the authors focus on the timing of oxygen concentrations, but they do not address the amplitude of these changes. Modern oxygen concentrations in the ETNA OMZ are $\sim 90 \mu\text{mol/kg}$ (2.01 ml/L). The highest oxygen concentrations are 182 $\mu\text{mol/kg}$ (4.08 ml/L). This is a change in oxygen of almost 100 $\mu\text{mol/kg}$ from the HS1 to the modern, which is extremely large!!! For comparison, the difference between HS1 and modern is -10 to 10 $\mu\text{mol/kg}$ in the Peru Margin (Erdem et al 2020, <https://doi.org/10.5194/bg-17-3165-2020>) and $<40 \mu\text{mol/kg}$ in the Arabian Sea (Lu et al, 2022, <https://doi.org/10.1016/j.gca.2022.06.001> and Costa et al., 2023, <https://doi.org/10.1016/j.gca.2022.12.022>). Are the mechanisms proposed in this paper able to cause such a large amplitude shift in oxygen concentrations? Is that oxygen change consistent with carbon storage/release during the deglaciation.

Finally, I was surprised that the manuscript did not consider the findings of Wang et al., 2024, "Global oceanic oxygenation controlled by the Southern Ocean through the last deglaciation", Science Advances. Their study presents whole ocean integrated oxygen concentrations during the last deglaciation, and they similarly find oxygenation events during HS1 and YD. However they have a different interpretation than present in this manuscript: the oxygen content of the ocean is not controlled by AMOC, but by circulation in the Southern Ocean. How do the findings of that paper influence the interpretation of the data presented here? I think it is worth some consideration and discussion.

%%% More specific comments

Lines 35-39: The evidence for AMOC slowing down as result of current and future climate change is highly debated. The text as written here misrepresents the field as having more consensus than it does, and the agreement amongst CMIP6 models is not conclusive given the difficulties of reconstructing AMOC within models. See, for example, "Challenges simulating the AMOC in climate models" by Jackson et al., 2023 <https://doi.org/10.1098/rsta.2022.0187>. Something like "there is debate about whether AMOC will slow down or not" is more realistic of our current understanding (or lack thereof!).

Lines 66-75: The authors should acknowledge that the description of AMOC variability in this paragraph is specifically derived from Pa/Th records. I am a fan of Pa/Th, but there are many who are not, and other proxies may report different deglacial AMOC variability, in amplitude, timing, or both. Better to be upfront about where these interpretations are coming from.

Results and discussion = Include errors when citing specific oxygen concentrations in the text. For example, if the text currently says "4.0 ml/l" it should be updated to say "4.0 ml/l (3.5-4.3 ml/l, 95% confidence interval)", or even just 3.5-4.3 ml/l (95% confidence interval)". This will assist readers in knowing when the variability in oxygen concentrations is actually significant.

Figure 2: Specify that this is a meridional SST gradient.

Figure 2d: The caption specifies the "shaded area corresponds to the 95% envelope" but there are 2 shaded areas on the figure. One is around the data points, that's fine. The other gray area could be the 95% envelope on the modern value, or it could be the range considered "low oxia". Please specify in the caption. Either way, it appears that none of the variability in the Holocene is significant.

Lines 140-141: Please explain how calcite preservation can be used as an indicator for remineralization.

Lines 153-176: I found this paragraph difficult to follow. It starts with meridional temperature gradients, then jumps to atmospheric circulation and wind stress, then to AMOC geometry (Line 157 "deep AMOC cell") and strength (Line 157 "... was weakened"), back to SST temperature gradients, back to winds. This text would be easier to follow if it were restructured. This could be by event, or mechanism, or something else, but it would strongly benefit from being rewritten for clarity.

Line 168: Please define NACW.

Lines 175-176: See above comment on Figure 2D about whether this event is actually significant.

Line 180-182: Are these subpolar foraminifera species planktonic? Since the rest of the manuscript is about benthic foraminifera, it should be specified if these lines are referring to planktonic instead.

Line 215: Does HS1 contain meltwater pulses? It is certainly associated with IRD deposition events, indicating the presence of widespread icebergs, but that is different from there being cold, fresh, liquid meltwater. See, for example, Bouttes et al, 2023, "Deglacial climate changes as forced by different ice sheet reconstructions", <https://doi.org/10.5194/cp-19-1027-2023>. I think the text could be modified by just removing the words "...and suggests there were two distinct meltwater pulses during the HS1..."

Lines 261-262: What is the reasoning for assuming Fe/Ca and Ti/Ca deposition at this site will be coincident with Heinrich events occurring in the North Atlantic? Considering the abundance of radiocarbon dates, it is not clear to me why making this assumption and imposing it on the age model is necessary.

Lines 278-280: The sentence "The studied... (Mulitza et al., 2005) is not complete.

Lines 290-309: Some of this information may be useful in the main text when restructuring Lines 153-176 for clarity.

Lines 362: How was the cutoff of 2.6 ka chosen for the Late Holocene? Generally I have seen the Late Holocene cutoff at ~4.2 ka. I suspect the younger cutoff age was selected in order to exclude the "high oxygen event" centered at ~5 ka. However this approach requires the preconception of a significant event at 5 ka before the significance has actually been calculated! Regardless, this seems like a strange way to calculate errors. Are there replicate samples for which reproducibility can be evaluated? Looking at the Kranner et al 2022, there is certainly a lot of scatter on their calibration from modern to reconstructed oxygen content in the global dataset (their Figure 3). They do not report errors on their regression parameters (shame on them). Proper use of this calibration should propagate errors from the calibration into the new data presented in this manuscript. It might be worth emailing Kranner et al to see if they have the error information that they could send them to you. Otherwise I would recommend re-running their calibration in order to get the proper errors so that they can be applied to your dataset.

(Remarks on code availability)

Version 1:

Reviewer comments:

Reviewer #2

(Remarks to the Author)

I have reviewed the new version of the manuscript and the responses to the suggestions and questions raised by the other three reviewers. I acknowledge the responses and changes made to the manuscript and the discussion added to improve the explanations of the potential mechanisms involved in the ventilation of the study area and the comparison with other records in the North Atlantic.

Regarding the responses to my comments, I agree with the majority of the key points and appreciate the effort to include more data and discuss in more detail the use and reliability of the EBFOI (Supplementary Information 4.1). What emerges from this exercise (Figure 4.3) is that reconstructing absolute values of oxygen from benthic foraminifera is very challenging and that different approaches can yield different values, making the approach more qualitative than quantitative, which is absolutely fine. The reconstructed oxygen curve now presented (Figure 2 main text) suggests that the site has always been more oxygenated than it is today, and during some periods was well outside the lower limb of the OMZ.

There are several aspects that still need to be considered. I explain them below:

Line 19, "low oxygen" (commented by Reviewer 1)

Line 61, "In this study, we used the Enhanced Benthic Foraminifera Oxygen Index (EBFOI, Kranner et al., 2022)", authors have used their updated database and EBFOI (Supplementary 4.1). I suggest to make a reference in the text to this updated compilation.

-Current oxygen levels at the core site and oxygen profiles.

Along the text there are several references to different oceanographic databases. I think it might be worth using and plotting a single data source where possible.

Figure 1b uses the weighted average-gridding, data from WOA 127 2018 (Boyer et al., 2018; García et al., 2019), and DIVA (line 127-128); the text refers to Lauvset et al., 2022 (line 86-87); Lauvset et al., 2016; Key et al., 2015) in line 178, 550-551; but supplementary information and Supplementary figure 4.1, uses data set GLODAP version 2.2023 (Lauvset et al., 2023).

-lines 102-104, suggest re-phrase as subtropical gyres are indicated twice in the text

- lines 169-173 and authors' reply to comment #2: "We agree that organic matter degradation within the sediment is an important control of oxygen levels at the seafloor/ in pore waters. However, this kind of degradation would leave its mark as calcite dissolution. Foraminiferal shells were very well preserved throughout the core and we hence see no evidence of for changes in remineralization (lines 169-173)."

I would provide an alternative explanation to lines 169-173 or remove the paragraph.

First, dissolution marks are difficult to assess without SEM-assisted views. Second, benthic foraminifera living in organic carbon-rich environments with pore water oxygen depletion calcify, and their shells are generally preserved in the fossil record without obvious signs of dissolution (judging from the plethora of published Pleistocene and Holocene records in highly productive regions off Africa and elsewhere, and from laboratory experiments). Third, organic carbon degrades in the seafloor, which can lead to a decrease in oxygen concentration in pore waters relative to bottom waters, even more so in a highly productive (mesotrophic to eutrophic) environment. The percentage of infaunal taxa in the core seems to support this, and the authors seem to agree with this as stated in the supplementary information (see updated TROX model and review in Glock, 2023, <https://doi.org/10.5194/bg-20-3423-2023>). Ruling out organic carbon degradation in the core and its potential influence on pore water oxygen concentrations is, I think, a big statement.

- Lines 114-115; 629-630, lines 203 and Supplementary information and authors' reply to Comment#2

In lines 114-115, 629-630 is indicated that EBFOI indicates bottom waters and pore waters then in line 203 it is mentioned only bottom waters and, in the supplementary information is not clear "the EBFOI is one of the best proxy tools that allows for a quantitative reconstruction of water oxygen concentration in the fossil record", and then there is the authors' reply to reviewer Comment#2 "As pointed out by the reviewer, stress species do show a combination of both effects, however the same cannot be said for the EBFOI used here. As shown by Kranner et al. (2022), the EBFOI constitutes a representation of the whole livable habitat which in turn should reflect bottom water oxygenation trends".

This is confusing. I would suggest that the authors follow what is stated in the original publications they referenced and be consistent in the main text and supplementary information about what kind of information the EBFOI index provides, even more so in the context of a mesotrophic to eutrophic environment as indicated for the core site.

I quote from Kranner et al., (2022): "Therefore, we enhanced the BFOI and introduce enhanced BFOI (EBFOI) formulas by using all available data benthic foraminifers provide, calculating the whole livable habitat of benthic foraminifers, including bottom water oxygenation (BWO) and pore water oxygenation (PWO)."

The same issue is again indicated in Schmiel et al. (2023)-study mentioned in the Supplementary Information. I quote: "In our application, we interpret the estimated dissolved oxygen values as a combined bottom- and pore-water oxygen (BPWO) signal and, thus, as a mixed signal of bottom-water ventilation and food fluxes."

-Line 201-202, check the sentence after Romero

-line 204, check units

-I suggest to round the oxygen values (not include decimals) when giving reconstructed oxygen values. I think is worthless to give decimals with the large uncertainties in the calibrations and assignments of fauna to oxygen categories.

-line 563-564, do the values reported for the oxygen content of ESACW represent "poorly oxygenated" waters?

-line 598, "um"

-Line 599, which geochemical analysis were performed on benthic foraminifera?

-Line 709: the link <https://zenodo.org/doi/10.5281/zenodo.10806183> delivers "page not found"

Comments on supplementary material

-Supplementary: "Data from Timm (1992) was also considered, however, was not included as the benthic foraminifera are reported in different units to the studies mentioned above."

I do not understand; data from Timm are in counts, which can be directly converted to percent.

-supplementary Fig. 4.3, revisit the text of the legend

-Supplementary 4: it is not clear why authors opted for using Kranner et al., 2022 instead the modification of Schmiel et al., 2023 after indicating "We find that this study and Schmiel et al. (2023) compilations using both calibrations show estimated values more approximated to the modern values, therefore the record and interpretations presented here are based on the dissolved oxygen concentrations estimated from our compilation using the Kranner et al. (2022) calibration." and "This contrasts the EBFOI and BWOx calculation using the data from Schmiel et al. (2023) and this study compilation, where the second peak is clearly represented. Furthermore, the high oxygen concentrations are even higher for the Last Glacial Maximum (LGM) and Younger Dryas (YD)."

In my opinion, the text is confusing regarding the choice of calibration and would benefit from clarification.

- “oxygen-depleted southern waters such as the Antarctic Intermediate Water”, in the supplementary text, is the AAIW attribution as oxygen-depleted correct?

“Secondly, our oxygenated periods coincide with increased upwelling (Figure 2c-d) related to stronger trade winds off NW Africa, suggesting higher nutrient content during these oxygenated times. In addition, the infaunal foraminifera content in our site (on average > 62%) indicates meso-eutrophic conditions (high organic matter concentrations) at the seafloor.”

I have constructed a graph using the % infaunal species (Figure 5) and the productivity indicators and oxygen (Figure 2). The figure does not show a systematic correlation between oxygen, productivity and infauna. The increase in organic matter inferred from the infauna (and shown in the figure) does not match the increase in opal/diatoms. On the other hand, the increase in infaunal species coincides with low reconstructed oxygen (influence of PWO + BWO?). Then I wonder if the opal and diatom record of GeoB7926-2, located 10 degrees north of GeoB9512-5, is representative of what is happening off Senegal.

Please consider these observations.

- References: Please, check whether these references are missing. I could not find them in the reference list of the tracked changes version of the manuscript

Tuchen et al., 2019; Poggemann et al., 2017; Poggeman et al., 2018; Rühlemann et al., 2004; Weldeab et al., 2016; Reissig et al., 2019; Holbourn et al., 2013, etc

Some additional references that authors might find useful for:

-Alternative approaches to assign benthic foraminifera species to oxygen affinities: see Tetard et al., 2024, <https://doi.org/10.1016/j.marmicro.2024.102380>.

-discussing the double peak in HS1, which has been reviewed in Hodell et al., 2017, <https://doi.org/10.1002/2016PA003028>

(Remarks on code availability)

Reviewer #3

(Remarks to the Author)

The authors have thoroughly answered all the questions that were raised in the review. I recommend the publication of the manuscript.

(Remarks on code availability)

Reviewer #4

(Remarks to the Author)

Thank you for your thorough response to my previous review. I just have a one item to follow up on.

The review response notes that the sites currently have 87.6 $\mu\text{mol/kg}$ oxygen, while sites to the north have 102.2 $\mu\text{mol/kg}$. Therefore if the NA STG shifted south, we might expect to see the oxygen content at the site to increase by ~15 $\mu\text{mol/kg}$. However, in the paleo record, oxygen content increases to values as high as 200 $\mu\text{mol/kg}$ in HS1 and YD, which is much higher than can be accounted for by movement of the NA STG. The response attributes the increase to "stronger winds ventilating the upper ocean", but can stronger winds account for near doubling of oxygen content? Are there some sensitivity tests that could be done to show that winds are capable of creating this amplitude of oxygen change (nearly 100 $\mu\text{mol/kg}$ increase)?

(Remarks on code availability)

Version 2:

Reviewer comments:

Reviewer #2

(Remarks to the Author)

I would like to thank the authors for their thorough responses to my review and their efforts to improve the manuscript.

Final remark: In the manuscript and the supplementary material, the key events B-A is listed, whereas it are not shown in any of the figures.

I recommend the manuscript for publication.

(Remarks on code availability)

Reviewer #4

(Remarks to the Author)

I do not understand the response linking high productivity to high dissolved oxygen. If there is high productivity, there will be abundant organic carbon, and respiration of that organic carbon will draw down oxygen. Upwelling does not ventilate the ocean, in the sense that it brings deep waters to the surface and not the other way around. Ventilation happens in regions of downwelling, in which water masses recently in contact with the atmosphere bring high oxygen to the deep waters. So this response does not make sense.

I still find the 200 mol/kg concentrations in the OMZ to be puzzling. Lu et al 2020 (EPSL) found oxygen concentrations that were less than 50 mol/kg in the OMZ. I feel like that paper should be cited somewhere in the manuscript, but the understanding the discrepancy between them is another can be done in future work. I do think it is important for the authors to acknowledge it though.

Regarding the switch from "BWO" to "BWPO", I think the text should use more careful phrasing so as not to mislead the reader. In the response to reviewer 2, the authors state "we do not rule out organic carbon degradation", but the way the text is currently written, the possibility of organic carbon degradation is not adequately presented. Here are some rephrasing recommendations:

L19 = "...we present a new deglacial high-resolution record of combined bottom and porewater oxygen..."

L22-23 = "...during the Younger Dryas (YD). These periods of high oxygen may correspond times of low organic carbon flux *. Alternatively they may be driven by steeper meridional..." If you feel there is a good argument to rule out the possibility or low organic carbon flux, then you can input that where the asterisk is

L76 = The Tuchen 2019 paper doesn't say anything about ventilation, it just talks about water mass transport. Looking at Figure 1, the high oxygen surface waters ~40°N barely make it 200 m down. The text in this Line either needs a different reference or the part about ventilation needs to be removed.

L78 = "To reconstruct the combined bottom and porewater oxygen (BPWO) variability..."

L88 = "The BPWO record presented here..."

L100 = "Here we present a new, high-resolution, and continuous record of BPWO..."

L117 onwards for the rest of the paper= say BPWO instead of just "oxygen". Otherwise this nuance will get lost, and readers will forget that it is combined bottom and porewater oxygen.

L150 = Figure 5 is called out before Figures 3 and 4. They should probably be rearranged.

(Remarks on code availability)

The original reviewer comments are in black font and the corresponding response is green font. The lines cited in the response correspond to the lines in the non-tracked changes version.

Reviewer #1 (Remarks to the Author):

1/1) Including a more in depth discussion of the link between AMOC and ventilation in this region. Are there records from other regions of the Atlantic that could test the author's assertion of ventilation changes?

In the text we now compare our data to nearby records - the intermediate depth Caribbean during the HS1 and YD (Schmiedl et al., 2023) and upper western tropical Atlantic during the YD (Lynch-Stiglitz et al. 2024). These records are now also plotted in Supplementary Information 5 (Figure S5.1). We also reference the compilation of Wang et al., (2024), the compilation of Skinner et al. (2021) and the studies of Slowey and Curry (1992) and Wharton et al. (2024). Line numbers are: LGM (line 228), HS1 (line 247) and YD (line 200).

These records support our conclusions that there is higher oxygen content in the upper subtropical North Atlantic during times of AMOC perturbation. One difference is that the high resolution GeoB9512-5 record shows two oxygen peaks in HS1 (line 248).

1/1a How about the mechanism in terms of northward gyre shift? Given the potential significance for future climate the mechanic piece linking oxygenation in the ETNA and AMOC strength is particularly important.

We cite Reissig et al. (2019) (lines 192-196) to show the relationship between northward gyre shifts with our oxygen record.

1/2 The deglacial Atlantic is probably one of the best sampled times and regions in paleoceanography. Thus, it's a bit of a missed opportunity that this paper is not more in conversation with other records. This should clearly include other regional paleo-oxygen records. For example, the authors mention in passing a contrast with deeper oxygenation (Lines 149-151). Exactly where and at what depth these records are important and should be discussed if not shown in a summary figure.

This comment is also addressed above **in 1/1**. We note that, although the deglacial Atlantic is well sampled compared to other regions, ventilation records at shallow /intermediate depths are rather sparse. Most reconstructions come from below 2,000 m water depth (Skinner et al., 2021; Wilson et al., 2014; Oliver et al. 2010). Also, some studies only focus on changes between the LGM and the Holocene and do not record the deglaciation (Slowey and Curry, 1992; Wharton et al., 2024).

1/3 Beyond oxygenation, though, both carbonate chemistry and temperature records from the ETNA and North Atlantic more broadly should be used to contextualize these findings.

We now cite studies that show warming in the Atlantic subsurface during AMOC perturbation and added further discussion of the role of temperature for oxygen concentrations (261-266) (thorough discussion is also provided in the last 2 paragraphs of Supplementary Information 5). A direct quantification of the effect of temperature on oxygen at GeoB9512-5 is, however, difficult since bottom water temperature reconstructions are sparse in the area and prone to relatively large errors.

The nearest published $\Delta\text{CO}_3\text{-2}$ record, which is that of Oppo et al. (2023) at 950 m in the west Atlantic shows low saturation, and low Cd, during HS1 and the YD, which they interpret as reduced AAIW at their site, and increased, better-ventilated, northern sourced waters during these intervals. This is consistent with our findings and is included in lines 198-200.

Minor things

1/4 - I'd strongly suggest the authors avoid the terms "low oxig" and "high oxig". Ocean oxygen research is already plagued by too many poorly defined or inconsistent terminology. Neither of these terms are in common use and introducing additional terminology here is not a service to the paper. As a quantitative oxygen record is generated, I'd recommend using actual quantitative values throughout. If the authors must use a qualitative term for oxygen range, then this range (not just the 3ml/L divide between them) needs to be defined in the body at first use and not in the methods.

corrected

- **Line 131:** "abundance of" corrected now line 149

- **Line 206:** are -> and line has been removed after revisions

Reviewer #2 (Remarks to the Author):

Comment #1: Oxygen categories, oxygen units and the OMZ of the Eastern Atlantic
Oxygen categories and the OMZ of the Eastern Atlantic: oxygen categories are not universal and thresholds vary between authors. This is commented in one of the papers cited in the text (Karstensen) but there is more literature discussing this (e.g., Paulmier and Ruiz-Pino, 2009; Holfmann et al., 2011; Levin, 2018). This is relevant in the context of this manuscript and I believe it deserves further explanation in the text. **Oxygen categories were removed.**

For example, the definition of OMZ provided by Levin (2018) as areas with oxygen values <22 micromol/kg (0.5 ml/l) excludes the study area as OMZ region. This has been mentioned in Karstensen et al. (2008) and Paulmier and Ruiz-Pino (2009). Therefore, lines 283-284 “modern low oxygen values of 2.01 ml/l are typical of the OMZ margin” and considerations along the text regarding the study area as “OMZ”, it would require some re-consideration/discussion. The value 2.01 ml/l is not even within the “suboxic range” indicated in line 344. For some authors it could be considered only mild hypoxia (Hofmann et al., 2011) or hypoxia (Levin, 2018). **Lines 60-66 now specifically refer to the conditions in the lower margin of the ETNA OMZ where the studied site is located. The definition of this area as an OMZ was taken from Brandt et al., 2015 and Karstensen et al., 2015, as ETNA OMZ core reaches minimal values of 40 umol/kg (line 62).**

Units: Following the information WOA18_oxygen (referenced by authors: <https://www.nodc.noaa.gov/OC5/WOD/wod18-notes.html>), oxygen units are provided micromol/kg. Most oceanographic literature discussing OMZ regions (e.g., Karstensen, Stramma, cited in the text) provide data in micromol/kg, so I wonder why authors are providing oceanographic sections in ml/l? **(Figure 1). We indeed provided the units in ml/l following the method of Kranner et al. (2022). We agree with the reviewer and convert all units to umol/kg (lines 405-415).**

Comment #2: Reconstruction of oxygen in bottom waters

I understand from the authors' text that the manuscript intends to reconstruct oxygen in bottom waters (lines 59-61, 199, 223). However, benthic foraminifera depend on oxygen concentrations in pore waters as well as food resources (I am not providing references as I understand the authors know them). Pore waters can have much lower oxygen

concentrations than bottom waters in mesotrophic and eutrophic environments. Given the high productivity of the study area (Figure 1), it would be reasonable to expect that the supply of organic carbon to the seafloor and its degradation in the sediment would reduce the oxygen concentration in the pore waters relative to the bottom waters.

We agree that organic matter degradation within the sediment is an important control of oxygen levels at the seafloor/ in pore waters. However, this kind of degradation would leave its mark as calcite dissolution. Foraminiferal shells were very well preserved throughout the core and we hence see no evidence of for changes in remineralization (lines 169-173).

Indeed, the background value of infaunal benthic foraminifera is high 40% (Figure 6). What is more, the use of "stress species" (line 350 "well adapted to oxygen depleted environments") would not be an indication of bottom water only, but of the combination of both, bottom water and pore water (and food resources). This would be even more important when the core is at the boundary of a non-OMZ region (see comment #1).

As pointed out by the reviewer, stress species do show a combination of both effects, however the same cannot be said for the EBFOI used here. As shown by Kranner et al. (2022), the EBFOI constitutes a representation of the whole livable habitat which in turn should reflect bottom water oxygenation trends. This is in part confirmed by the coherence of dissolved oxygen concentrations derived from EBFOI with the modern values studied by the cited authors, as well as, for our core tops in the studied area. We added clarification in the method lines 372-374.

(2/2c) I think it would be reasonable to try to obtain additional proxies that would allow the micropaleontological signals to be interpreted with greater confidence. Sedimentary geochemical proxies (Corg, U, Mn, etc.) would be one option to try to assess the influence of primary production and independent information about the oxygenation of the bottom waters.

We agree with the reviewer that it would be interesting to compare our record with geochemical proxies but feel that those proxies are not sufficiently understood to provide a robust interpretation. Redox sensitive metals, such as U and Mn, in bulk sediments or foraminifera, give a signal which combines the oxygen signal of bottom water and that of porewaters, meaning bottom water oxygen can only be isolated if the rain rate of organic carbon can be constrained (Hoogakker et al., 2024-in review). Organic carbon input in the

area is known to have varied through time (Figure 2; Romero et al., 2008; Zarries et al., 2010; Bradtmiller et al., 2016) but given the concurrent changes in oxygen (and thus possible changes in remineralization in the water column) input is not well constrained.

Most of the oxygen proxies perform well at relatively low-oxygen concentrations (Hoogakker et al., 2024-in review), that occur in extremely depleted pacific OMZs which have oxygen concentrations close to 0 $\mu\text{mol}/\text{kg}$ (e.g. Karstensen et al., 2008). Our studied area is located in in ETNA OMZ (that reaches minima of 40 $\mu\text{mol}/\text{mol}$ at 420 m depth) and the studied site GeoB9512-5 (793 m depth) is surrounded by more even oxygenated conditions (87.6 $\mu\text{mol}/\text{kg}$; Lauvset et al., 2022) where benthic foraminifera associations are the ideal proxy to monitor changes in oxygen concentration (e.g. Kaiho, 1994; Kranner et al., 2022).

For example, lines 137-139 refer to some previously published primary productivity inferences, but they are not in the same core and they are not plotted alongside the core being studied.

We now include the data of Romero et al. (2008) discussed in lines 233-238 and in Figure 2c-d.

(2/2e) I would like to mention that, as discussed in Kranner et al. (2022), EBFOI intends to "include bottom water oxygenation (BWO) and pore water oxygenation (PWO)". The index does not refer to BWO and the oceanographic context does not allow such a direct interpretation (see comment #3).

See also the response to Comment No. 3 below. Although the index refers to the oxygen concentrations of the whole livable habitat, the transfer function results show a good approximation with measured dissolved oxygen concentrations of bottom waters (Kranner et al., 2022). To address this point thoroughly, we used 15 core top data (Barragán-Montilla, 2024) from the tropical Atlantic to create a local calibration of the relationship between oxygen and BF assemblages. This is described Supplementary Information 4.1. However, we found that the approach of Kranner is more suitable for our study, as it is based on a geographically wider selection of sites that covers a larger range of oxygen values.

Comment #3: The use of EBFOI as a proxy

The EBF0I index (Kranner et al., 2022) is an improved version of the Kaiho BFOI index (1994, 1999). The index has several limitations which are not mentioned in the text. I think those limitations require some reflection. We now discuss this in detail in Supplementary Information 4.1.

EBFOI and BFOI are based on the relative contribution of taxa stacked into oxygen categories. This means that certain species are assumed to live only in the dysoxic, suboxic, low oxic and high oxygen categories (ranges are given in the manuscript). Kranner et al. (2022) increased the number of samples from Kaiho's original 80 to 270. Despite the increase in the number of samples, this does not appear to be a large number for a worldwide ecological distribution. The original data do not follow FAIR principles, so it is not possible to reproduce or evaluate the context by which a species is assigned to an oxygen category. As far as I can tell from the papers cited above, there is no disclosure of the number of samples, type of assemblage (live, dead), ranges of variability, or optima of tolerance used to assign a species to a particular oxygen range or the impact of different size fraction in the index calculation.

We agree with the reviewer, as the way oxic categories for Benthic Foraminifera species oxygen preferences were defined is unfortunately not fully disclosed in the paper by Kranner et al. (2022). However, the conditions reported for each species by the authors agree with observations made by previous studies. Furthermore, we also estimated the EBF0I with the compilation provided by Schmiedl et al. (2023) that was specifically focused on the Atlantic Ocean, which mostly agrees with the compilation of Kranner (Supplementary Information 4.2, sheet 6). In addition, we put together a compilation of living foraminifera data in PANGAEA specifically for the eastern tropical Atlantic (which was not considered in the two previous compilations) and we could assign almost the same oxygen categories reported by Kranner et al. (2022) and Schmiedl et al. (2023). Our compilation did, however, improve the data used in the EBF0I. This data is summarized in Supplementary Information 4.2, sheet 6.

For example, after applying the index, the supplementary data presented in the manuscript include assignments of oxygen levels to individuals identified to genus level (genus sp.), dubious assignments; "aff", individuals of the order Lagenina (oolina, lagenina, lavidentalina) or rare species. I wonder about the ecological/statistical basis for assigning individuals of which we do not know the species, or cosmopolitan "lagenids", rare species, to an oxygen category? Another consideration is the maximum oxygen value

obtained from the index which is around 5ml/l. If we convert ml/l to micromol/kg (<https://shorturl.at/fluGU>), we get 220 micromol/kg, which is roughly the oxygen value of bottom waters below 1250 metres (https://www.ewoce.org/gallery/A6_OXYGEN.gif). Does the value obtained from the index make sense in the context of the current/past oceanographic context (see comment #1)?

We agree with the reviewer that there is still room for improvement in this particular proxy (EBFOI), as with any other proxy, but we would like to point out that the available compilation from Kranner et al. (2022) is robust and shows good agreement modern and Cenozoic values (See Fig 3 in <https://doi.org/10.1038/s41598-022-05295-8>). The reconstructed modern oxygen concentration based on counts from a multicorer (GeoB9512-4) sediment surface at our station shows good agreement with the modern annual mean dissolved oxygen concentrations (lines 428-431) and provides robust evidence that the reconstructions based on EBFOI are reliable enough to interpret the fossil record in our site. With respect to the “dubious” assignments, particularly those made to genus level, we find that the assignment of an oxygen category would not disprove the EBFOI, species identification of some of these genera (*Lagena*, *Oolina*, *Fissurina* etc.) to species levels is extremely difficult due to the lack of proper holotype availability and the extremely high benthic foraminifera diversity. In the case of the dubious determination such as with some species of *Bolivina*, *Pyrgo* and *Quinqueloculina*, we can infer from the existing literature (e.g. Murray, 1991, 2006; Schmiedl et al., 2023) that the environmental conditions have been well defined to a genus level and are therefore applicable.

The dataset: <https://doi.pangaea.de/10.1594/PANGAEA.962951> (can only comment on the metadata, see comment below on data availability below), include, as far as understand, unpublished core tops. I wonder why authors have “reported microhabitats and oxygen conditions for most species were extracted from various references including Murray (1991), Murray (2006) Kaiho (1994) and Kranner et al. (2022) among others” instead of using their own regional compilation and interpretation. For example, they could include not only their new core tops but also previously published datasets [I found for example: Haake, F W (1980): Relative abundance of dead benthic foraminifera in East Atlantic surface sediments off Senegal and Gambia (Table 2). PANGAEA, <https://doi.org/10.1594/PANGAEA.536233> also in Senegal, is this is not relevant?] and perhaps additional datasets from the west coast of Africa. They could compare

distributions with bottom water oxygen values, with organic carbon and any other relevant environmental variables available to extract their own interpretation, which might fit or might not with the results of EBF0I. I view that suggestion as a reasonable exercise not only to support their interpretation instead of relying on previously regional data and/or interpretations from distant areas. The interpretation of benthic foraminifera as proxies for food and oxygen concentrations are not written in stone; they should be based on a reasonable set of data that allow for quantitative relations, and should be continuously updated as benthic foraminifera ecological information increases.

This is a valid appreciation, however, the species reported here agree with the oxygen preferences assigned by Kranner et al. (2022) and Kaiho (1994). We have revised these preferences with the compilation of Schmiedl et al., 2023 and our own of living benthic foraminifera assemblages in the tropical eastern Atlantic (see response above to Reviewer 2, comment 2e) in Supplementary Information 4.1 and 4.2. To demonstrate the sensitivity to the choice of the calibration data set we calculated the EBF0I and dissolved oxygen concentration with the compilation of (1) Kranner; (2) Schmiedl; and (3) a combination of these two with the available data from living species of the eastern tropical Atlantic (Figure S 4.4, in the Supplementary Information 4.1). The underestimation of oxygen mentioned by Schmiedl et al. (2023) is evident, but a marked improvement is obtained when using the Schmiedl et al. (2023) data in combination with our updated compilation. The record produced using the new core top compilation is very similar to that produced from the combination of Kranner and Kaiho oxyc preference data used in the first submission of this manuscript. It does not change any of our conclusions.

It is not the aim of our study to evaluate the EBF0I proxy, however, we find the points raised by the reviewer valid and relevant, and we have added a detailed discussion on Supplementary Information 4.1, as we have improved the oxygen preference compilation of species by adding new data from the tropical eastern Atlantic (Supplementary Information 4.2, sheets 1 and 6). We can conclude that the interpretation is not affected substantially by the proxy's limitations, and we used in the final version the updated benthic foraminifera oxygen preferences data in the transfer function of Kranner et al (2022) as it would more easily allow future comparisons with other records from different ocean basins.

Other comments

Taxonomic concepts: I would like to mention that I really appreciate that authors provide images and taxonomic concepts of the identified foraminifera. By scrolling down the list of species (supplementary table), some species designations call my attention. It appears that the taxonomy is not fully updated, as it is indicated in line 335. These are random examples (I have not checked all). We thank the author for this appreciation, and we also want to commend the reviewer for taking the time to revise the taxonomy in detail.

Virgulina subquamosa, unaccepted: link; *Nonion fabum*, unaccepted: link; *Eggerelloides scaber?*, not recognized, *Eggerelloides scabrum* instead? link

I suggest authors access WoRMS using “worrsm” package in R (Chamberlain and Vanhoorne, 2023) to get quick access to updated taxonomy. We have reviewed and updated the taxonomy in Supplementary Information No. 3 to the current state, as for July 16, 2024.

Transport:

Have the authors considered the influence of downslope transport in their assemblages?.

E. scabrum, *C. williamsoni*, *E. macellum* are generally “shallow water species” and *B. spathulata/dilatata/subaeranensis* and *N. fabum* are common in “shelf” environments.

I think it would be worth evaluating transport and consequently, do not include transported species in interpretations. This is a very valid point; however, these species are known to have wider depth ranges, therefore they are not necessarily strictly confined to a certain depth. For example, there is evidence of shallow to bathyal distribution of *Elphidium macellum* (Holbourn et al., 2013), or *B. subaenariensis* (Holbourn et al., 2013; Jones, 1994; Parker, 1954; Phleger & Parker, 1951). Also, the preservation state does not suggest transport from shallower depths. We stress that the removal of the species mentioned by the reviewer does not substantially change the results (see figure below).

Datasets for review:

It is impossible to review and comment on datasets (benthic foraminifera, including oxygen assignments, age model), because they are not available, they are password protected. I would like to notice that, if requested, Pangaea facilitates temporary access to dataset for confidential reviewing. I have tried to convert the supplementary file with data counts to excel but the format of the table is unfriendly and it takes time to arrange columns properly for data wrangle. The files will remain protected for general access until publication of the manuscript. As far as I tested, the data set was accessible via the links provided for reviewer access. This was via the code ocean capsule service provided by the journal. The data was uploaded as a PDF due to the submission system. We will make the data available in Excel for the reviewers and apologize for the inconvenience.

Age model:

Lines 257-261 and 272-273 Age model: Please clarify “median calendar ages are calculated from uncalibrated Fe/Ca and Ti/Ca” and “Median calendar ages were also calculated for sedimentary and uncalibrated”. I understand authors are using Fe/Ca peaks in their record (that have been previously shown to function as good proxies for Heinrich events) to gather additional “tie points” for age model construction. I am sorry, but I do not understand how Fe/Ca is transferred to “calibrated” years? The calendar ages for the XRF data were modeled using BACON with the radiocarbon ages measured at the site. It good be useful to see how the inclusion of Fe/Ca tie points fits with the radiocarbon dates/calibrated ages, in particular during HS1 when both data are available (figure 4). See comment “Datasets for review”. What we did was see when HS1 occurred in our record, we did not model ages with additional tie points. This data and the code used for the age model were provided via the code ocean service from the journal, we understood this capsule was available for reviewers. More details are included in Reviewer’s 4 lines 261-262 comment.

Paleoclimate and paleoceanographic discussion

The discussion of lines 153-176 might benefit from a cartoon or similar depicting the paleoceanographic scenarios mentioned in the text. We considered a cartoon but feel that it might fail to represent the complexity of the involved processes.

Lines 165-167: Previous studies in the area indicated that enhanced easterly winds during the YD and HS1 enhanced primary production and organic carbon flux to the sea floor (Bradtmiller et al., 2016), Is then possible to conciliate enhanced productivity with increased ventilation of the thermocline? Please refer to comment #2. **Yes, we observe increased ventilation during periods of increased productivity during HS1 and the YD recorded along the NW African coast (Romero et al., 2008; Bradtmiller et al., 2016). The corresponding text has been rewritten (lines 233-237) to clarify this important point.**

There is evidence from modeling experiments that AMOC perturbation leads to stronger winter trade winds (e.g., deMenocal and Rind, 1993). Since both the strength of the North Atlantic subtropical cell and coastal upwelling are driven by trade winds, it is reasonable that the upwelling-driven productivity increases with the ventilation of the thermocline.

Figures:

- **Figure 1:** shadowing obscures evaluating the vertical oxygen distribution and it does not allow to see the contouring. I suggest providing data following international oceanography convention (see comment #1) without shadowing. Once contours are provided, readers can visually identify areas of low oxygen and its value. **We removed the shadow oxygen categories.**
- **Figure 2:** how the gradient has been constructed in b) (any age interpolation has been used to subtract a record from other)? **Yes, the data was interpolated, we have updated this in the figures, we added this clarification in lines 209.**
- **Figure 3:** label of y-axis in panel a) does not match Figure caption (BWOx versus Fisher alpha). Line in c) does not match figure caption (dissolved oxygen?). **Corrected**
- Perhaps figure 2 and figure 3 could be merged into one. **Figure 3 has been removed in this revision.**
- **Figure 5:** I have similar considerations to Figure 1. Colouring and shadowing pre-conditions the readers to authors' water masses identifications. Perhaps arrows, lines, etc it would be sufficient. **The figure was modified accordingly, now figure 4.**
- **Line 51:** "hard-shelled": consider that there are foraminifera with organic and quitinous walls when referring to foraminifera. **Corrected.**

References:

Please, re-visit reference list; the first entry has not author, sometimes journals are abbreviated and some others they are not, some references are with et al. and others are not. **References were revised.**

Reviewer #3 (Remarks to the Author):

I see two minor weaknesses in the argumentation of the manuscript:

1. The authors use the Pa/Th record from the Bermuda rise as reference for AMOC strength. Though this record is strikingly matching the presented new results from the ETNA-OMZ, care needs to be taken with this Pa/Th record as a record for AMOC strength. The latter currently is debated to potentially be overprinted by other effects than AMOC strength. Here, more careful writing and/or addition of other proxies than Pa/Th could help to improve the manuscript. **Added in lines 90-93.**

2. The argument that increasing ENACW strength and additionally increased ventilation of ENACW causes the oxygen increase during cold periods is conclusive. Yet the authors leave out the possibility of increased ventilation and contribution of the southern sourced waters WSACW and ESACW. This probably could be added in 1 or 2 sentences. **We agree with the reviewer that the observed changes in oxygen concentration can be associated with changes in the relative contribution of northern and southern sourced central waters. This point is specifically addressed in line (lines 190-192).**

Further Minor suggestions:

Please check the consistent use of the abbreviation ETNA OMZ sometimes OMZ or ETNA is missing. **Corrected.**

Response to minor comments in the PDF not addressed:

It is very interesting to see that this peak is situated after the onset of the early Holocene.

You may want to compare not only the short term data to Repschläger et al., 2015 but also to the subsurface transport data shown in Repschläger et al., 2017 fig. 3d

Repschläger, J., Garbe-Schönberg, D., Weinelt, M. & Schneider, R. (2017) Holocene evolution of the North Atlantic subsurface transport. *Clim. Past* 13, 333-344, doi:10.5194/cp-13-333-2017 (2017).

These observations about cold subsurface waters, and lack of NAC, at the Azores Front in the early Holocene are interesting and probably would relate to more polar water reaching our site. In the end, we did not include direct comparison of our data with that of Repschläger (2015, 2017) in the manuscript, preferring to focus on the main mechanism, strengthened winds increasing ventilation within the gyre.

please add evidence for AMC changes from a proxy other than Pa/Th, as the latter is debated controversially. We agree that Pa/Th is not a perfect AMOC proxy. However, the deglacial AMOC changes indicated by Pa/Th from the Bermuda Rise are corroborated by numerous independent observations (e.g. bipolar seesaw, decline in benthic $\delta^{13}\text{C}$) and are hence robust. In lines 91-95 we now mention potential problems with this proxy.

Reviewer #4 (Remarks to the Author):

First, throughout the manuscript, oxygen concentrations are reported as ml/l. This is likely a result of the original relationships established between benthic foraminiferal assemblages and oxygen as presented by, e.g., Kaiho 1994. However, the modern convention is to report oxygen concentrations in $\mu\text{mol}/\text{kg}$, as exemplified by data products like WOA18, GLODAPv2, and WOCE all using $\mu\text{mol}/\text{kg}$. It is rather tedious for the reader to be constantly doing the conversion throughout the text. Other papers on benthic foram assemblages, like Erdem 2019 and 2020, are starting to make the switch to these more modern preferred units, and I recommend the authors of this manuscript do so as well. Using $\mu\text{mol}/\text{kg}$ will also make it much easier to compare these results with those published using other proxies (e.g., the $\delta^{13}\text{C}$ of Hoogakker et al., 2015, 2018).

Corrected, see response to Reviewer #2's first comment "units".

Next, the authors focus on the timing of oxygen concentrations, but they do not address the amplitude of these changes. Modern oxygen concentrations in the ETNA OMZ are $\sim 90 \mu\text{mol}/\text{kg}$ (2.01 ml/L). The highest oxygen concentrations are $182 \mu\text{mol}/\text{kg}$ (4.08 ml/L). This is a change in oxygen of almost $100 \mu\text{mol}/\text{kg}$ from the HS1 to the modern, which is extremely large!!! For comparison, the difference between HS1 and modern is -10 to $10 \mu\text{mol}/\text{kg}$ in the Peru Margin (Erdem et al 2020, <https://doi.org/10.5194/bg-17-3165-2020>) and $<40 \mu\text{mol}/\text{kg}$ in the Arabian Sea (Lu et al, 2022, <https://doi.org/10.1016/j.gca.2022.06.001> and Costa et al., 2023, <https://doi.org/10.1016/j.gca.2022.12.022>). Are the mechanisms proposed in this paper

able to cause such a large amplitude shift in oxygen concentrations? North Atlantic oxygen gradients are greater than those observed in Peru. Oxygen concentrations at our site at 15° N are currently 87.6 $\mu\text{mol/kg}$ while, to the north at 20° N they are 102.2 $\mu\text{mol/kg}$. A southward shift of the NA STG of a few degrees would expose the site to better ventilated waters. The main mechanism of ventilation though, is stronger winds ventilating the upper ocean.

Is that oxygen change consistent with carbon storage/release during the deglaciation.

(This comment is related to Reviewer 1 comment 2 above.)

The upper Atlantic represents only a small part of the global ocean. The global picture of dissolved oxygen and atmospheric CO₂ is dominated by Southern Ocean overturning and ventilation (Jaccard et al., 2016; Burke and Robinson, 2012; Wang et al., 2024). The GeoB9512 record shows both similarities and differences to the global pattern. Unlike the global record, 9512 shows a decrease in oxygen (from around 92 to 42 $\mu\text{mol/kg}$) between LGM and the Holocene, so it does not appear to contribute to the rise in atm CO₂. Rising oxygen in GeoB9512 coincides with rising atm CO₂, during the HS1 and the YD. These findings are consistent with other records from the North Atlantic which suggest that the upper Atlantic is better ventilated during times of AMOC slowdown, while $\delta^{13}\text{C}$ and $\delta^{14}\text{C}$ indicate the deep Atlantic is less well ventilated (Skinner et al., 2021), due to less convection and stronger carbon pump (Yu et al., 2019).

Considering global contributions to the carbon budget from one site would always be difficult. From our data in GeoB9512-5, it would be particularly uncertain, as there are large changes in both productivity and ventilation concurrently. For instance, increased productivity along the African coast in HS1 is registered by research in the area (Bouimetarhan et al., 2013; Bradtmiller et al., 2016; Romero et al., 2008), which at our site is a high oxygen interval.

Finally, I was surprised that the manuscript did not consider the findings of Wang et al., 2024, "Global oceanic oxygenation controlled by the Southern Ocean through the last deglaciation", Science Advances. Their study presents whole ocean integrated oxygen concentrations during the last deglaciation, and they similarly find oxygenation events during HS1 and YD.

However they have a different interpretation than present in this manuscript: the oxygen content of the ocean is not controlled by AMOC, but by circulation in the Southern Ocean. How do the findings of that paper influence the interpretation of the data presented here? I think it is worth some consideration and discussion.

The relationship between oxygen and AMOC strength in the upper Atlantic in this record does not mean that AABW formation is not the dominant control of oxygen in the global ocean (Wang et al., 2024), and we do not suggest this. AMOC / NADW formation is of course related to AABW formation, but this is slightly removed from what we can conclude from our study.

We understand the importance of putting our results into context with previous analyses, therefore we added a discussion of our results with other paleo-oxygenation records of the Atlantic with similar age resolution in the manuscript (lines 225-229; and 243-245, Supplementary Information 5).

In addition, our inferences are also supported by other studies. For example, Chang et al. 2008 (<https://www.nature.com/articles/ngeo218>) found that in times of AMOC slowdown, the equatorial flow of the Subtropical Cell is strengthened and acts like a closed cell, transporting warmer and saltier waters into the equatorial zone. The warming, that has been seen in the eastern (Weldeab et al., 2016) and western (Came et al., 2007; Oppo et al., 2023; Poggeman et al., 2018; Rühlemann et al., 2004) Atlantic (Supplementary Information 5) and oxygenation observed in our study, would be consistent with higher subsurface renewal rates also from enhanced subtropical gyre circulation. In addition, given the modern control in ETNA OMZ (*Oceanographic Setting* in the main manuscript), we believe the hypothesis presented in our manuscript is more compatible with the southern position of the subtropical gyres (Portilho-Ramos et al., 2017; Pinho et al., 2021; Reißig et al., 2019; Repschläger et al., 2015), the evidence suggesting enhanced eastern winds (Romero et al., 2008; Bradtmiller et al., 2016), and the enhanced ventilation of the North Atlantic Subtropical Gyre during the LGM (Slowey and Curry, 1992; Wharton et al., 2024 - <https://www.nature.com/articles/s41586-024-07655-y>).

%%% More specific comments

- **Lines 35-39:** The evidence for AMOC slowing down as result of current and future climate change is highly debated. The text as written here misrepresents the field

as having more consensus than it does, and the agreement amongst CMIP6 models is not conclusive given the difficulties of reconstructing AMOC within models. See, for example, “Challenges simulating the AMOC in climate models” by Jackson et al., 2023 <https://doi.org/10.1098/rsta.2022.0187>. Something like “there is debate about whether AMOC will slow down or not” is more realistic of our current understanding (or lack thereof!). **Added in lines 38-40, 90-93.**

- **Lines 66-75:** The authors should acknowledge that the description of AMOC variability in this paragraph is specifically derived from Pa/Th records. I am a fan of Pa/Th, but there are many who are not, and other proxies may report different deglacial AMOC variability, in amplitude, timing, or both. Better to be upfront about where these interpretations are coming from. **Corrected, now lines 86-90.**
- **Results and discussion =** Include errors when citing specific oxygen concentrations in the text. For example, if the text currently says “4.0 ml/l” it should be updated to say “4.0 ml/l (3.5-4.3 ml/l, 95% confidence interval)”, or even just “3.5-4.3 ml/l (95% confidence interval)”. This will assist readers in knowing when the variability in oxygen concentrations is actually significant. **Corrected.**
- **Figure 2:** Specify that this is a meridional SST gradient. **Corrected.**
- **Figure 2d:** The caption specifies the “shaded area corresponds to the 95% envelope” but there are 2 shaded areas on the figure. One is around the data points, that’s fine. The other gray area could be the 95% envelope on the modern value, or it could be the range considered “low oxig”. Please specify in the caption. Either way, it appears that none of the variability in the Holocene is significant. **Corrected.**
- **Lines 140-141:** Please explain how calcite preservation can be used as an indicator for remineralization. **Added now lines 169-173.**
- **Lines 153-176:** I found this paragraph difficult to follow. It starts with meridional temperature gradients, then jumps to atmospheric circulation and wind stress, then to AMOC geometry (Line 157 “deep AMOC cell”) and strength (Line 157 “... was weakened”), back to SST temperature gradients, back to winds. This text would be easier to follow if it were restructured. This could be by event, or mechanism, or something else, but it would strongly benefit from being rewritten for clarity. **The text has been restructured.**
- **Line 168:** Please define NACW. **Corrected, line 187.**
- **Lines 175-176:** See above comment on Figure 2D about whether this event is actually significant. **The values are now reported with the associated error.**

- **Line 180-182:** Are these subpolar foraminifera species planktonic? Since the rest of the manuscript is about benthic foraminifera, it should be specified if these lines are referring to planktonic instead. *This text was removed after the revision.*
- **Line 215:** Does HS1 contain meltwater pulses? It is certainly associated with IRD deposition events, indicating the presence of widespread icebergs, but that is different from there being cold, fresh, liquid meltwater. See, for example, Bouttes et al, 2023, “Deglacial climate changes as forced by different ice sheet reconstructions”, <https://doi.org/10.5194/cp-19-1027-2023>. I think the text could be modified by just removing the words “..and suggests there were two distinct meltwater pulses during the HS1...” *This paragraph has been rewritten, see lines 243-258.*
- **Lines 261-262:** What is the reasoning for assuming Fe/Ca and Ti/Ca deposition at this site will be coincident with Heinrich events occurring in the North Atlantic? Considering the abundance of radiocarbon dates, it is not clear to me why making this assumption and imposing it on the age model is necessary. *Our formulation of the age modelling methods was indeed misleading. Our age model is only based on radiocarbon ages of planktic foraminifera. Since radiocarbon might be influenced by changes in reservoir ages, we used Fe/Ca and Ti/Ca ratios to test and support the age model. Previous work has shown that Heinrich Stadials are associated with increased dust input and elevated Fe/Ca and Ti/Ca in sediments off NW Africa (Mulitza et al. 2008). In our core, Heinrich Stadials 1 and 2 are indeed associated with peaks in Fe/Ca and Ti/Ca which further supports our chronology. We have re-arranged the wording to avoid confusion.*
- **Lines 278-280:** The sentence “The studied... (Mulitza et al., 2005) is not complete. *Corrected.*
- **Lines 290-309:** Some of this information may be useful in the main text when restructuring Lines 153-176 for clarity. *Corrected, now lines 185-187.*
- **Lines 362:** How was the cutoff of 2.6 ka chosen for the Late Holocene? Generally I have seen the Late Holocene cutoff at ~4.2 ka. I suspect the younger cutoff age was selected in order to exclude the “high oxygen event” centered at ~5 ka. However this approach requires the preconception of a significant event at 5 ka before the significance has actually been calculated! Regardless, this seems like a strange way to calculate errors. Are there replicate samples for which reproducibility can be evaluated? Looking at the Kranner et al 2022, there is certainly a lot of scatter on their calibration from modern to reconstructed oxygen content in the global

dataset (their Figure 3). They do not report errors on their regression parameters (shame on them). Proper use of this calibration should propagate errors from the calibration into the new data presented in this manuscript. It might be worth emailing Kranner et al to see if they have the error information that they could send them to you. Otherwise, I would recommend re-running their calibration in order to get the proper errors so that they can be applied to your dataset. We are not using the date of 2.6 ka as a definition for the late Holocene. The errors used in our study are obtained from the standard deviation of the EBF0I-calculated oxygen concentrations from coretops and recent samples. We chose the period of the last 2.6 kyrs to include samples in relatively stable AMOC conditions, as we have shown oxygen changes are related to AMOC strength variations, and there is evidence suggesting AMOC potentially declined at around 5 ka BP, and even if this is not inclusive, we prefer to avoid this bias. The oxygenation data showed a normal distribution; therefore, the error was propagated with the age uncertainty. In R Studio, the code has also been updated. This is explained in lines 405-417.

References cited in this response

- Barragán-Montilla, S.: Benthic Foraminifera counts off NW Africa during the last deglaciation, <https://doi.org/10.1594/PANGAEA.962951>, 2024.
- Bouimetarhan, I., Groeneveld, J., Dupont, L., and Zonneveld, K.: Low- to high-productivity pattern within Heinrich Stadial 1: Inferences from dinoflagellate cyst records off Senegal, *Global and Planetary Change*, 106, 64–76, <https://doi.org/10.1016/j.gloplacha.2013.03.007>, 2013.
- Bradtmiller, L. I., McGee, D., Awalt, M., Evers, J., Yerxa, H., Kinsley, C. W., and deMenocal, P. B.: Changes in biological productivity along the northwest African margin over the past 20,000 years: AFRICAN MARGIN PALEOPRODUCTIVITY, *Paleoceanography*, 31, 185–202, <https://doi.org/10.1002/2015PA002862>, 2016.
- Brandt, P., Bange, H. W., Banyte, D., Dengler, M., Didwischus, S.-H., Fischer, T., Greatbatch, R. J., Hahn, J., Kanzow, T., Karstensen, J., Körtzinger, A., Krahnemann, G., Schmidtko, S., Stramma, L., Tanhua, T., and Visbeck, M.: On the role of circulation and mixing in the ventilation of oxygen minimum zones with a focus on the eastern tropical North Atlantic, *Biogeosciences*, 12, 489–512, <https://doi.org/10.5194/bg-12-489-2015>, 2015.

- Burke, A. and Robinson, L. F.: The Southern Ocean's Role in Carbon Exchange During the Last Deglaciation, *Science*, 335, 557–561, <https://doi.org/10.1126/science.1208163>, 2012.
- Came, R. E., Curry, W. B., Oppo, D. W., Broccoli, A. J., Stouffer, R. J., and Lynch-Stieglitz, J.: North Atlantic intermediate depth variability during the Younger Dryas: Evidence from benthic foraminiferal Mg/Ca and the GFDL R30 Coupled Climate Model, in: *Geophysical Monograph Series*, vol. 173, edited by: Schmittner, A., Chiang, J. C. H., and Hemming, S. R., American Geophysical Union, Washington, D. C., 247–263, <https://doi.org/10.1029/173GM16>, 2007.
- deMenocal, P. B. and Rind, D.: Sensitivity of Asian and African climate to variations in seasonal insolation, glacial ice cover, sea surface temperature, and Asian orography, *J. Geophys. Res.*, 98, 7265–7287, <https://doi.org/10.1029/92JD02924>, 1993.
- Holbourn, A. E. L. and Henderson, A. S.: *Atlas of benthic foraminifera*, Natural History Museum, Chichester, West Sussex ; Hoboken, NJ, 2013.
- Hoogakker, B., Davis, C., Wang, Y., Kusch, S., Nilsson-Kerr, K., Hardisty, D., Jacobel, A., Reyes Macaya, D., Glock, N., Ni, S., Sepúlveda, J., Ren, A., Auderset, A., Hess, A., Meissner, K., Cardich, J., Anderson, R., Barras, C., Basak, C., Bradbury, H., Brinkmann, I., Castillo, A., Cook, M., Costa, K., Choquel, C., Diz, P., Donnenfield, J., Elling, F., Erdem, Z., Filipsson, H., Garrido, S., Gottschalk, J., Govindankutty Menon, A., Groeneveld, J., Hallman, C., Hendy, I., Hennekam, R., Lu, W., Lynch-Stieglitz, J., Matos, L., Martínez-García, A., Molina, G., Muñoz, P., Moretti, S., Morford, J., Nuber, S., Radionovskaya, S., Raven, M., Somes, C., Studer, A., Tachikawa, K., Tapia, R., Tetard, M., Vollmer, T., Wu, S., Zhang, Y., Zheng, X.-Y., and Zhou, Y.: Reviews and syntheses: Review of proxies for low-oxygen paleoceanographic reconstructions, <https://doi.org/10.5194/egusphere-2023-2981>, 9 January 2024.
- Jaccard, S. L., Galbraith, E. D., Martínez-García, A., and Anderson, R. F.: Covariation of deep Southern Ocean oxygenation and atmospheric CO₂ through the last ice age, *Nature*, 530, 207–210, <https://doi.org/10.1038/nature16514>, 2016.
- Jones, R. W. and Brady, H. B.: *The Challenger foraminifera*, Oxford University Press, Oxford ; New York, 149 pp., 1994.

- Kaiho, K.: Benthic foraminiferal dissolved-oxygen index and dissolved-oxygen levels in the modern ocean, *Geol*, 22, 719, [https://doi.org/10.1130/0091-7613\(1994\)022<0719:BFDOIA>2.3.CO;2](https://doi.org/10.1130/0091-7613(1994)022<0719:BFDOIA>2.3.CO;2), 1994.
- Karstensen, J., Stramma, L., and Visbeck, M.: Oxygen minimum zones in the eastern tropical Atlantic and Pacific oceans, *Progress in Oceanography*, 77, 331–350, <https://doi.org/10.1016/j.pocean.2007.05.009>, 2008.
- Karstensen, J., Fiedler, B., Schütte, F., Brandt, P., Körtzinger, A., Fischer, G., Zantopp, R., Hahn, J., Visbeck, M., and Wallace, D.: Open ocean dead zones in the tropical North Atlantic Ocean, *Biogeosciences*, 12, 2597–2605, <https://doi.org/10.5194/bg-12-2597-2015>, 2015.
- Kranner, M., Harzhauser, M., Beer, C., Auer, G., and Piller, W. E.: Calculating dissolved marine oxygen values based on an enhanced Benthic Foraminifera Oxygen Index, *Sci Rep*, 12, 1376, <https://doi.org/10.1038/s41598-022-05295-8>, 2022.
- Lauvset, S. K., Lange, N., Tanhua, T., Bittig, H. C., Olsen, A., Kozyr, A., Alin, S., Álvarez, M., Azetsu-Scott, K., Barbero, L., Becker, S., Brown, P. J., Carter, B. R., Da Cunha, L. C., Feely, R. A., Hoppema, M., Humphreys, M. P., Ishii, M., Jeansson, E., Jiang, L.-Q., Jones, S. D., Lo Monaco, C., Murata, A., Müller, J. D., Pérez, F. F., Pfeil, B., Schirnick, C., Steinfeldt, R., Suzuki, T., Tilbrook, B., Ulfsbo, A., Velo, A., Woosley, R. J., and Key, R. M.: GLODAPv2.2022: the latest version of the global interior ocean biogeochemical data product, *Earth Syst. Sci. Data*, 14, 5543–5572, <https://doi.org/10.5194/essd-14-5543-2022>, 2022.
- Lynch-Stieglitz, J., Vollmer, T. D., Valley, S. G., Blackmon, E., Gu, S., and Marchitto, T. M.: A diminished North Atlantic nutrient stream during Younger Dryas climate reversal, *Science*, 384, 693–696, <https://doi.org/10.1126/science.adi5543>, 2024.
- Murray, J. W.: *Ecology and palaeoecology of benthic foraminifera*, Longman scientific and technical copublished in the United States with John Wiley and sons, Essex New York, 1991.
- Murray, J. W.: *Ecology and Applications of Benthic Foraminifera*, 1st ed., Cambridge University Press, <https://doi.org/10.1017/CBO9780511535529>, 2006.
- Oliver, K. I. C., Hoogakker, B. A. A., Crowhurst, S., Henderson, G. M., Rickaby, R. E. M., Edwards, N. R., and Elderfield, H.: A synthesis of marine sediment core $\delta^{13}\text{C}$ data over the last 150 000 years, *Clim. Past*, 6, 645–673, <https://doi.org/10.5194/cp-6-645-2010>, 2010.

- Oppo, D. W., Lu, W., Huang, K. -F., Umling, N. E., Guo, W., Yu, J., Curry, W. B., Marchitto, T. M., and Wang, S.: Deglacial Temperature and Carbonate Saturation State Variability in the Tropical Atlantic at Antarctic Intermediate Water Depths, *Paleoceanog and Paleoclimatol*, 38, e2023PA004674, <https://doi.org/10.1029/2023PA004674>, 2023.
- Parker, F. L.: Distribution of the foraminifera in the north-eastern Gulf of Mexico, *Bulletin of the Museum of Comparative Zoology Harvard*, 111, 453–588, 1954.
- Phleger, F. B. and Parker, F. L.: PART II. FORAMINIFERA SPECIES, in: *Geological Society of America Memoirs*, vol. 46, Geological Society of America, 1–80, <https://doi.org/10.1130/MEM46-p2-0001>, 1951.
- Pinho, T. M. L., Chiessi, C. M., Portilho-Ramos, R. C., Campos, M. C., Crivellari, S., Nascimento, R. A., Albuquerque, A. L. S., Bahr, A., and Mulitza, S.: Meridional changes in the South Atlantic Subtropical Gyre during Heinrich Stadials, *Sci Rep*, 11, 9419, <https://doi.org/10.1038/s41598-021-88817-0>, 2021.
- Poggemann, D. -W., Nürnberg, D., Hathorne, E. C., Frank, M., Rath, W., Reißig, S., and Bahr, A.: Deglacial Heat Uptake by the Southern Ocean and Rapid Northward Redistribution Via Antarctic Intermediate Water, *Paleoceanog and Paleoclimatol*, 33, 1292–1305, <https://doi.org/10.1029/2017PA003284>, 2018.
- Portilho-Ramos, R. C., Chiessi, C. M., Zhang, Y., Mulitza, S., Kucera, M., Siccha, M., Prange, M., and Paul, A.: Coupling of equatorial Atlantic surface stratification to glacial shifts in the tropical rainbelt, *Sci Rep*, 7, 1561, <https://doi.org/10.1038/s41598-017-01629-z>, 2017.
- Reißig, S., Nürnberg, D., Bahr, A., Poggemann, D. -W., and Hoffmann, J.: Southward Displacement of the North Atlantic Subtropical Gyre Circulation System During North Atlantic Cold Spells, *Paleoceanog and Paleoclimatol*, 34, 866–885, <https://doi.org/10.1029/2018PA003376>, 2019.
- Repschläger, J., Weinelt, M., Kinkel, H., Andersen, N., Garbe-Schönberg, D., and Schwab, C.: Response of the subtropical North Atlantic surface hydrography on deglacial and Holocene AMOC changes, *Paleoceanography*, 30, 456–476, <https://doi.org/10.1002/2014PA002637>, 2015.
- Repschläger, J., Garbe-Schönberg, D., Weinelt, M., and Schneider, R.: Holocene evolution of the North Atlantic subsurface transport, *Clim. Past*, 13, 333–344, <https://doi.org/10.5194/cp-13-333-2017>, 2017.

- Romero, O. E., Kim, J., and Donner, B.: Submillennial-to-millennial variability of diatom production off Mauritania, NW Africa, during the last glacial cycle, *Paleoceanography*, 23, 2008PA001601, <https://doi.org/10.1029/2008PA001601>, 2008.
- Rühlemann, C., Mulitza, S., Lohmann, G., Paul, A., Prange, M., and Wefer, G.: Intermediate depth warming in the tropical Atlantic related to weakened thermohaline circulation: Combining paleoclimate data and modeling results for the last deglaciation, *Paleoceanography*, 19, 2003PA000948, <https://doi.org/10.1029/2003PA000948>, 2004.
- Schmiedl, G., Milker, Y., and Mackensen, A.: Climate forcing of regional deep-sea biodiversity documented by benthic foraminifera, *Earth-Science Reviews*, 244, 104540, <https://doi.org/10.1016/j.earscirev.2023.104540>, 2023.
- Skinner, L. C., Shackleton, N. J., and Elderfield, H.: Millennial-scale variability of deep-water temperature and $\delta^{18}\text{O}_{\text{dw}}$ indicating deep-water source variations in the Northeast Atlantic, 0–34 cal. ka BP, *Geochim Geophys Geosyst*, 4, 2003GC000585, <https://doi.org/10.1029/2003GC000585>, 2003.
- Slowey, N. C. and Curry, W. B.: Enhanced ventilation of the North Atlantic subtropical gyre thermocline during the last glaciation, *Nature*, 358, 665–668, <https://doi.org/10.1038/358665a0>, 1992.
- Wang, Y., Costa, K. M., Lu, W., Hines, S. K. V., and Nielsen, S. G.: Global oceanic oxygenation controlled by the Southern Ocean through the last deglaciation, *Sci. Adv.*, 10, eadk2506, <https://doi.org/10.1126/sciadv.adk2506>, 2024.
- Weldeab, S., Friedrich, T., Timmermann, A., and Schneider, R. R.: Strong middepth warming and weak radiocarbon imprints in the equatorial Atlantic during Heinrich 1 and Younger Dryas, *Paleoceanography*, 31, 1070–1082, <https://doi.org/10.1002/2016PA002957>, 2016.
- Wharton, J. H., Renoult, M., Gebbie, G., Keigwin, L. D., Marchitto, T. M., Maslin, M. A., Oppo, D. W., and Thornalley, D. J. R.: Deeper and stronger North Atlantic Gyre during the Last Glacial Maximum, *Nature*, <https://doi.org/10.1038/s41586-024-07655-y>, 2024.
- Wilson, D. J., Crocket, K. C., Van De Flierdt, T., Robinson, L. F., and Adkins, J. F.: Dynamic intermediate ocean circulation in the North Atlantic during Heinrich Stadial 1: A radiocarbon and neodymium isotope perspective, *Paleoceanography*, 29, 1072–1093, <https://doi.org/10.1002/2014PA002674>, 2014.

- Yu, J., Menviel, L., Jin, Z. D., Thornalley, D. J. R., Foster, G. L., Rohling, E. J., McCave, I. N., McManus, J. F., Dai, Y., Ren, H., He, F., Zhang, F., Chen, P. J., and Roberts, A. P.: More efficient North Atlantic carbon pump during the Last Glacial Maximum, *Nat Commun*, 10, 2170, <https://doi.org/10.1038/s41467-019-10028-z>, 2019.
- Zarriess, M. and Mackensen, A.: The tropical rainbelt and productivity changes off northwest Africa: A 31,000-year high-resolution record, *Marine Micropaleontology*, 76, 76–91, <https://doi.org/10.1016/j.marmicro.2010.06.001>, 2010.

Responses are in blue fonts and lines refer to the tracked changes files.

Reviewer #2 (Remarks to the Author):

I have reviewed the new version of the manuscript and the responses to the suggestions and questions raised by the other three reviewers. I acknowledge the responses and changes made to the manuscript and the discussion added to improve the explanations of the potential mechanisms involved in the ventilation of the study area and the comparison with other records in the North Atlantic.

Regarding the responses to my comments, I agree with the majority of the key points and appreciate the effort to include more data and discuss in more detail the use and reliability of the EBFOI (Supplementary Information 4.1). What emerges from this exercise (Figure 4.3) is that reconstructing absolute values of oxygen from benthic foraminifera is very challenging and that different approaches can yield different values, making the approach more qualitative than quantitative, which is absolutely fine. The reconstructed oxygen curve now presented (Figure 2 main text) suggests that the site has always been more oxygenated than it is today, and during some periods was well outside the lower limb of the OMZ.

There are several aspects that still need to be considered. I explain them below:

Line 19, “low oxygen” (commented by Reviewer 1): removed

Line 61, “In this study, we used the Enhanced Benthic Foraminifera Oxygen Index (EBFOI, Kranner et al., 2022)”, authors have used their updated database and EBFOI (Supplementary 4.1). I suggest to make a reference in the text to this updated compilation.

Now line 57, text added to reference all the sources used in our database of bf preferences:

“In this study, we used the Enhanced Benthic Foraminifera Oxygen Index (EBFOI, Kranner et al., 2022), supplemented with additional data on benthic foraminifera oxygen preferences (Kaiho, 1994; Schmiedl et al., 2023; Haake, 1980; Licari and Mackensen, 2006; Lutze, 1980; details in Supplementary Material 4.1 and 4.2).”

-Current oxygen levels at the core site and oxygen profiles.

Along the text there are several references to different oceanographic databases. I think it might be worth using and plotting a single data source where possible. The text and figures have been updated to reference GLODAP v2 2022 (Lauvset et al., 2022) consistently. The updated line numbers are given in brackets below.

Figure 1b uses the weighted average-gridding, data from WOA 127 2018 (Boyer et al., 2018; García et al., 2019), and DIVA (line 127-128); the text refers to Lauvset et al., 2022 (line 86-87); (now line 80) Lauvset et al., 2016; Key et al., 2015) in line 178, (now line 142) line 550-551; (now line 302) but supplementary information and Supplementary figure 4.1, uses data set GLODAP version 2.2023 (Lauvset et al., 2023): Database reference has also been updated in lines 300, 398 and 415

-lines 102-104 suggest re-phrase as subtropical gyres are indicated twice in the text: STG is now mentioned only once (there was a related direct repetition in line 298, now deleted)

- lines 169-173 (Results)

and authors' reply to comment #2: "We agree that organic matter degradation within the sediment is an important control of oxygen levels at the seafloor/ in pore waters. However, this kind of degradation would leave its mark as calcite dissolution. Foraminiferal shells were very well preserved throughout the core and we hence see no evidence of for changes in remineralization (lines 169-173)." I would provide an alternative explanation to lines 169-173 or remove the paragraph.

(Now lines 162-164) we want to state that calcite preservation appeared good throughout the core. Shortened / rephrased as follows:

„Although acids released by remineralization of organic matter under oxic conditions can result in calcite dissolution in seafloor sediments (Emerson and Archer, 1981; Emerson and Archer, 1990), foraminifera tests appeared well preserved throughout the core.”

First, dissolution marks are difficult to assess without SEM-assisted views. Second, benthic foraminifera living in organic carbon-rich environments with pore water oxygen depletion calcify, and their shells are generally preserved in the fossil record without obvious signs of dissolution (judging from the plethora of published Pleistocene and Holocene records in highly productive regions off Africa and elsewhere, and from laboratory experiments).

Planktic foraminifera are sensitive to dissolution to various degrees (Be, 1975) and it is usually possible to get a feel for preservation state of a sample from looking at the whole assemblage. High organic matter plus oxygen can result in very poor preservation of forams. In this core, calcite preservation appeared good throughout with at least no obvious intervals of poor preservation.

Third, organic carbon degrades in the seafloor, which can lead to a decrease in oxygen concentration in pore waters relative to bottom waters, even more so in a highly productive (mesotrophic to eutrophic) environment. The percentage of infaunal taxa in the core seems to support this, and the authors seem to agree with this as stated in the supplementary information (see updated TROX model and review in Glock, 2023, <https://doi.org/10.5194/bg-20-3423-2023>). Ruling out organic carbon degradation in the core and its potential influence on pore water oxygen concentrations is, I think, a big statement.

We do not rule out organic carbon degradation - either in the water column or at the sea floor – and its role on oxygen consumption. We wish to say there were no obvious changes in preservation throughout the core, despite changes in BPWO and in organic matter input. Text has been updated to clarify (lines 162-164)

- Lines 114-115; 629-630; lines 203 and Supplementary information and authors' reply to Comment#2. In lines 114-115, (now 113) 629-630 (now ~ line 377) is indicated that EBF0I indicates bottom waters and pore waters then in line 203 (Fig 2 caption) it is mentioned only bottom waters and, in the supplementary information is not clear "the EBF0I is one of the best proxy tools that allows for a quantitative reconstruction of water oxygen concentration in the fossil record", (Sup Mat line 36) and then there is the authors' reply to reviewer Comment#2 "As pointed out by the reviewer, stress species do show a combination of both effects, however the same cannot be said for the EBF0I used here. As shown by Kranner et al. (2022), the EBF0I constitutes a representation of the whole livable habitat which in turn should reflect bottom water oxygenation trends".

This is confusing. I would suggest that the authors follow what is stated in the original publications they referenced and be consistent in the main text and supplementary information about what kind of information the EBF0I index provides, even more so in the context of a mesotrophic to eutrophic environment as indicated for the core site.

I quote from Kranner et al., (2022): “Therefore, we enhanced the BFOI and introduce enhanced BFOI (EBFOI) formulas by using all available data benthic foraminifers provide, calculating the whole livable habitat of benthic foraminifers, including bottom water oxygenation (BWO) and pore water oxygenation (PWO).”

The same issue is again indicated in Schmiel et al. (2023)-study mentioned in the Supplementary Information. I quote: “In our application, we interpret the estimated dissolved oxygen values as a combined bottom- and pore-water oxygen (BPWO) signal and, thus, as a mixed signal of bottom-water ventilation and food fluxes.”

In this study we follow the approach of Kranner et al., (2022) (as stated in line 59) and are reconstructing combined bottom- and pore- water oxygen. We now use the term BPWO consistently throughout the text. Line 113 (beginning of the Results section) updated to BPWO

(Line 364 deleted as repetitive)

Fig 2 Caption, updated to BPWO

Fig 5 Caption, updated to BPWO

Supplementary info 4.1 line 37, updated to BPWO

Sup Mat Fig 4.2 caption updated to BPWO

Sup Mat Fig S4.3 caption updated to BPWO

Sup Mat Fig S4.4 caption updated to BPWO

-Line 201-202 (Fig 2 Caption), check the sentence after Romero

-Line 204 (Fig 2 caption), check units

Caption now updated is both to use BPWO and $\mu\text{mol}/\text{kg}$

-I suggest to round the oxygen values (not include decimals) when giving reconstructed oxygen values. I think is worthless to give decimals with the large uncertainties in the calibrations and assignments of fauna to oxygen categories: Modified

-line 563-564, do the values reported for the oxygen content of ESACW represent “poorly oxygenated” waters?

Now line 314, Agree this is confusing, it refers to the values at the source, and not very relevant for the site. Values removed.

-line 598, “ μm ”: Corrected to μm

-Line 599 (now 346), which geochemical analysis were performed on benthic foraminifera?: Corrected

-Line 709: the link <https://zenodo.org/doi/10.5281/zenodo.10806183> delivers “page not found”: The link is updated in lines 454, 458 and 469

Comments on supplementary material

-Supplementary: “Data from Timm (1992) was also considered, however, was not included as the benthic foraminifera are reported in different units to the studies mentioned above.”

I do not understand; data from Timm are in counts, which can be directly converted to percent.

The taxonomic categories are different, for this reason the data of Timm (1992) from 48 sites from the Gulf of Guinea was not used. Since the data is never referred to, the cited sentence in Sup Mat 4 has been removed.

-supplementary Fig. 4.3, revisit the text of the legend

Figure S4.3 and its caption has been updated.

In the previous version we considered making our own calibration using the data from this study and discussed this briefly before deciding to use the calibration of Kanner et al., (2022) for consistency with other studies. In this new version of the text we only consider the calibration of Kanner et al., (2022) since that is the one used. Fig 4.3 has been simplified as a result to compare only the effects of different bf oxygen preference compilations.

-Supplementary 4: it is not clear why authors opted for using Kranner et al., 2022 instead the modification of Schmiedl et al., 2023 after indicating “We find that this study and Schmiedl et al. (2023) compilations using both calibrations show estimated values more approximated to the modern values, therefore the record and interpretations presented here are based on the dissolved oxygen concentrations estimated from our compilation using the Kranner et al. (2022) calibration.”

We opted to use our revised compilation, that includes data from Kaiho (1994), Kranner et al. (2022) and Schmiedl et al. (2023) as well as additional data compiled by us from Haake, 1980, Licari and Mackensen, 2006, Lutze, 1980 (Supplementary Information 4.3) as this is the most comprehensive collection of bf preference data Fig S4.3 shows that the compilation of Schmiedl (2023) and that of “this study” gives rather similar results for 15 Atlantic coretops. We use the calibration of Kranner et al (2022) to be consistent with other studies and because we are following their approach.

This has been rephrased lines 42-44 and 103-104

and “This contrasts the EBFOI and BWOx calculation using the data from Schmiedl et al. (2023) and this study compilation, where the second peak is clearly represented. Furthermore, the high oxygen concentrations are even higher for the Last Glacial Maximum (LGM) and Younger Dryas (YD).” (Sup Mat line 114)

In my opinion, the text is confusing regarding the choice of calibration and would benefit from clarification.

The text in this section has been extensively altered to clarify the points raised by the reviewer. The three most important changes are:

- The relationship between EBFOI of the 15 Atlantic core-tops and oxygen is now not referred to as a calibration, as we do not use it that way.
- Using the oxygen preference compilation of Schmiedl et al. (2023) or that of this study give rather similar results for the 15 Atlantic core-tops (Fig S4.3). We chose to use the compilation of this study because it has additional data for the east Atlantic. In order to be consistent with other studies, we used the calibration equations presented by Kranner et al., (2022). Text has been updated to clarify (Sup Mat 4, line 103-104)
- The discussion of the BPWO reconstructions of GeoB9512 data (starting Sup Mat line 110; Fig S4.4) has been expanded.

- “oxygen-depleted southern waters such as the Antarctic Intermediate Water”, in the

supplementary text, is the AAIW attribution as oxygen-depleted correct?: This has been removed

-“Secondly, our oxygenated periods coincide with increased upwelling (Figure 2c-d) related to stronger trade winds off NW Africa, suggesting higher nutrient content during these oxygenated times. In addition, the infaunal foraminifera content in our site (on average > 62%) indicates meso-eutrophic conditions (high organic matter concentrations) at the seafloor.” (Sup Mat, now line 204)

I have constructed a graph using the % infaunal species (Figure 5) and the productivity indicators and oxygen (Figure 2). The figure does not show a systematic correlation between oxygen, productivity and infauna. The increase in organic matter inferred from the infauna (and shown in the figure) does not match the increase in opal/diatoms. On the other hand, the increase in infaunal species coincides with low reconstructed oxygen (influence of PWO + BWO?).

The percentage of infaunal alone might not be the best way to judge productivity changes for the GeoB9512-5 record. If oxygen and organic matter input increase together, as at this site, then infaunal % may drop just because more benthic foraminifera (BF) who prefer high oxygen can be sustained, and there is an increase in BF overall. % infaunal BF and BPWO are anticorrelated in the record, but infaunal BF are abundant through the whole core.

Then I wonder if the opal and diatom record of GeoB7926-2 located 10 degrees north of GeoB9512-5, is representative of what is happening off Senegal.

Core GeoB7926-2 is at 20° 13'N. GeoB9512-5 is at 15° 20' N. The GeoB7926-2 opal record (Figure 2) is one of several records which show increased productivity along the West African coast during HS1 and the YD, attributed to stronger NE tradewinds increasing wind driven upwelling (main text line 240), and was plotted as an example of what is going on in the wider area. A study of four cores between 19 and 31°N along Western Africa, shows increased opal flux during HS1 and YD (Bradtmitter et al., 2016). Increased productivity was also recorded at sites GeoB9526 and GeoB9527 (12°26' N, 18°W) during HS2 and HS1 (Zarries and Mackenson, 2010). This reference has been added Sup Mat 4 line 204.

These two factors - the multiple records suggesting enhanced productivity during HS1 and YD in the area, and the mesotrophic to eutrophic conditions indicated by infaunal BF throughout the core, suggest that increases in oxygen during HS1 and YD recorded in GeoB9512 were not due to reduced oxygen consumption because of reduced productivity. We conclude that the increase in oxygen came from better ventilation by winds during these periods.

The text is clarified in Sup Mat 4, starting line 204.

Please consider these observations.

- References: Please, check whether these references are missing. I could not find them in the reference list of the tracked changes version of the manuscript

Tuchen et al., 2019; Poggemann et al., 2017; Poggeman et al., 2018; Rühlemann et al., 2004; Weldeab et al., 2016; Reissig et al., 2019; Holbourn et al., 2013, etc. The references have been revised.

Some additional references that authors might find useful for:

-Alternative approaches to assign benthic foraminifera species to oxygen affinities: see

Tetard et al., 2024, <https://doi.org/10.1016/j.marmicro.2024.102380>). This does look like a great resource for future studies

-discussing the double peak in HS1, which has been reviewed in Hodell et al., 2017, <https://doi.org/10.1002/2016PA003028>. Reference added in text, line 254 in the manuscript

Reviewer #3 (Remarks to the Author):

The authors have thoroughly answered all the questions that were raised in the review. I recommend the publication of the manuscript.

Reviewer #4 (Remarks to the Author):

Thank you for your thorough response to my previous review. I just have a one item to follow up on.

The review response notes that the sites currently have 87.6 $\mu\text{mol/kg}$ oxygen, while sites to the north have 102.2 $\mu\text{mol/kg}$. Therefore, if the NA STG shifted south, we might expect to see the oxygen content at the site to increase by ~ 15 $\mu\text{mol/kg}$. However, in the paleo record, oxygen content increases to values as high as 200 $\mu\text{mol/kg}$ in HS1 and YD, which is much higher than can be accounted for by movement of the NA STG.

The response attributes the increase to "stronger winds ventilating the upper ocean", but can stronger winds account for near doubling of oxygen content? Are there some sensitivity tests that could be done to show that winds are capable of creating this amplitude of oxygen change (nearly 100 $\mu\text{mol/kg}$ increase)?

We agree that it would be interesting and useful to see the response of modelled oxygen concentrations to deglacial changes in wind strength, but we are not aware of any published modelling results in this direction. However, the reconstructed increase in oxygen concentration to values as high as 200 $\mu\text{mol/kg}$ is in the range of modern observed oxygen concentrations along the African coast shown in Figure 1b. Values approaching 200 $\mu\text{mol/kg}$ are common in the area north of 30°N, where year-round upwelling leads to highest levels of biological activity. The high subsurface oxygen concentrations in these areas of highest productivity suggest that upper ocean ventilation can indeed dominate the distribution of dissolved oxygen in the modern ocean and is a plausible process to explain past variations in oxygen concentration.

The responses to the comments correspond to the bold blue text. The lines refer to the tracked changes file.

REVIEWER COMMENTS

Reviewer #2 (Remarks to the Author):

I would like to thank the authors for their thorough responses to my review and their efforts to improve the manuscript. Final remark: In the manuscript and the supplementary material, the key events B-A is listed, whereas it are not shown in any of the figures. **Thanks for catching this, Figures 2, 3, 5, S4.4 and S5.1 were updated.**

I recommend the manuscript for publication.

Reviewer #4 (Remarks to the Author):

I do not understand the response linking high productivity to high dissolved oxygen. If there is high productivity, there will be abundant organic carbon, and respiration of that organic carbon will draw down oxygen. Upwelling does not ventilate the ocean, in the sense that it brings deep waters to the surface and not the other way around. Ventilation happens in regions of downwelling, in which water masses recently in contact with the atmosphere bring high oxygen to the deep waters. So this response does not make sense.

I still find the 200 $\mu\text{mol}/\text{kg}$ concentrations in the OMZ to be puzzling. Lu et al 2020 (EPSL) found oxygen concentrations that were less than 50 $\mu\text{mol}/\text{kg}$ in the OMZ. I feel like that paper should be cited somewhere in the manuscript, but the understanding the discrepancy between them is another can be done in future work. I do think it is important for the authors to acknowledge it though.

We agree with the reviewer. We added Lu et al. (2020) in line 246 and added a clarification in line 248 (tracked changes file). I want to clarify that we do not suggest a direct link between high productivity and high dissolved oxygen. We agree with the mechanisms as the reviewer describes here. In the text we say we find higher dissolved oxygen even during periods where higher productivity has been recorded in the area, such as during the YD (lines from 241). BPWO is higher not because of, but despite, productivity being higher.

Regarding the switch from "BWO" to "BWPO", I think the text should use more careful phrasing so as not to mislead the reader. In the response to reviewer 2, the authors state "we do not rule out organic carbon degradation", but the way the text is currently written, the possibility of organic carbon degradation is not adequately presented. Here are some rephrasing recommendations: L19 = "...we present a new deglacial high-resolution record of combined bottom and porewater oxygen..." **We added "combined bottom and pore water" in parenthesis.**

L22-23 = "...during the Younger Dryas (YD). These periods of high oxygen may correspond times of low organic carbon flux *. Alternatively they may be driven by steeper meridional..." If you feel there is a good argument to rule out the possibility or low organic carbon flux, then you can input that where the asterisk is **We believe the way the text is written summarizes our findings, as we mentioned in previous responses, we did not make any organic carbon reconstructions therefore we prefer to exclude it in the summary.**

L76 = The Tuchen 2019 paper doesn't say anything about ventilation, it just talks about water mass transport. Looking at Figure 1, the high oxygen surface waters ~40°N barely make it 200 m down. The text in this Line either needs a different reference or the part about ventilation needs to be removed. **Removed.**

L78 = "To reconstruct the combined bottom and porewater oxygen (BPWO) variability..." **Accepted and modified.**

L88 = "The BPWO record presented here..." **Accepted and modified.**

L100 = "Here we present a new, high-resolution, and continuous record of BPWO..." **Accepted and modified.**

L117 onwards for the rest of the paper= say BPWO instead of just "oxygen". Otherwise this nuance will get lost, and readers will forget that it is combined bottom and porewater oxygen. **The modifications have been made accordingly.**

L150 = Figure 5 is called out before Figures 3 and 4. They should probably be rearranged. **The figures were re-arranged and numeration was modified accordingly.**